# Genomic insights into local adaptation and future climate-induced vulnerability of a keystone forest tree in East Asia

Yupeng Sang [1,3], Zhiqin Long [1,3], Xuming Dan [1], Jiajun Feng [1], Tingting Shi[1], Changfu Jia [1], Xinxin Zhang[1], Qiang Lai [1], Guanglei Yang [1], Hongying Zhang [1], Xiaoting Xu[1], Huanhuan Liu [1], Yuanzhong Jiang [1], Pär K. Ingvarsson [2], Jianquan Liu [1] ✉, Kangshan Mao [1] ✉ & Jing Wang [1] ✉

Rapid global climate change is posing a substantial threat to biodiversity. The assessment of population vulnerability and adaptive capacity under climate change is crucial for informing conservation and mitigation strategies. Here we generate a chromosome-scale genome assembly and re-sequence genomes of 230 individuals collected from 24 populations for *Populus koreana*, a pioneer and keystone tree species in temperate forests of East Asia. We integrate population genomics and environmental variables to reveal a set of climate-associated single-nucleotide polymorphisms, insertion/deletions and structural variations, especially numerous adaptive non-coding variants distributed across the genome. We incorporate these variants into an environmental modeling scheme to predict a highly spatiotemporal shift of this species in response to future climate change. We further identify the most vulnerable populations that need conservation priority and many candidate genes and variants that may be useful for forest tree breeding with special aims. Our findings highlight the importance of integrating genomic and environmental data to predict adaptive capacity of a key forest to rapid climate change in the future.

Climate change is predicted to become a major threat to biodiversity, and there is ample evidence of climate-induced local extinctions among plant and animal species[1]. To escape demographic collapses from threats, species must trace suitable locations or adjust to changing environments via phenotypic plasticity or innovation from standing genetic variation and de novo mutations[2]. However, migration, in order to keep pace with rapid climate change, may be difficult for many species[3]. Therefore, understanding and quantifying the evolutionary potential of a species for future adaptation is not only relevant for understanding whether and how natural species can persist in the context of climate change, but it can also benefit conservation and management strategies to cope with global biodiversity loss[4,5].

The spatial predictions of the effects of future climate change have largely relied on the knowledge of adaptive genetic variations on current climate conditions[6]. The traditional way to identify and quantify local adaptation via reciprocal transplant and/or common garden experiments[7–9] is challenging and often unfeasible for many wild non-model organisms due to experimental intractability, long generation times, or other challenges to obtaining fitness-related phenotypic traits[10]. With the advance of genomic technologies, it is now becoming increasingly affordable to generate population

[1]Key Laboratory for Bio-Resources and Eco-Environment, College of Life Sciences, Sichuan University, Chengdu, China. [2]Department of Plant Biology, Linnean Centre for Plant Biology, Swedish University of Agricultural Sciences, Uppsala, Sweden. [3]These authors contributed equally: Yupeng Sang, Zhiqin Long. ✉e-mail: liujq@nwipb.cas.cn; maokangshan@163.com; wangjing2019@scu.edu.cn

genomic data that can serve as a complementary strategy to examine local adaptation throughout the distribution range of one targeted species[11–13]. Although there may be millions of variants across the genome within any specific species, only some of these variants are expected to be related to climate adaptation[9,14]. The process of identifying the climate-associated genetic variation is not only critical for solving fine-scale patterns of local adaptation but also facilitates a more mechanistic understanding of how species respond to climate change[15]. Genotype–environmental association approaches are increasingly used to identify loci involved in climate adaptation[16]. Once candidates for locally adaptive allelic variation have been identified, it is possible to measure genomic offset, which assesses the amount of change in the genetic composition of a population that is required to track future environmental conditions[17,18]. As such, using genomic tools goes beyond modeling species range changes over time and can provide novel insights into assessing the evolutionary adaptive potential and predicting the disruption of local adaptation and species vulnerability in a changing climate[4,17,19,20].

Forest trees are efficient carbon sinks and play an increasingly leading role in the global carbon cycle and in combating climate change and global warming[10,21]. However, they are particularly vulnerable to maladaptation owing to their long generation times, for which climate change is likely to happen within the lifetimes of single individuals and making them especially challenged in terms of adapting to keep up with rapid climate change[22]. In this context, integrating genomic data into predictive models aimed at quantifying and mapping spatial patterns of climate maladaptation is especially important for long-lived tree species[5,23,24]. In this study, we accomplish such a goal for *Populus koreana* (Salicaceae), one

dominant tree species in temperate deciduous forests in East Asia. We assemble the first, to our knowledge, de novo chromosome-scale reference genome for this species and further re-sequence genomes of 230 individuals from 24 natural populations across its distribution range. We characterize genome-wide variations, including single-nucleotide polymorphisms (SNPs), small insertions/deletions (indels) and large structural variants (SVs). Based on these genomic datasets, we aim to: (1) infer the spatial patterns of genetic diversity, population structure, and evolutionary history; (2) dissect the genomic underpinnings of climate adaptation and investigate how past selection shaped patterns of adaptive allele frequencies across distributional ranges; and (3) quantify and map the vulnerable populations under future climate change.

## Results
### Chromosome-scale genome assembly of *P. koreana*
For de novo assembly of the *P. koreana* genome, we integrated data from three sequencing and assembly technologies: ~42.42 Gb of Nanopore long-read sequencing (106×), ~29.82 Gb of short-read Illumina sequencing (74×), and ~54.22 Gb of Hi-C paired-end reads (137×) (Supplementary Tables 1–4). The final assembly captured 401.4 Mb of the genome sequence, with contig N50 of 6.41 Mb and ~99.6% (~399.94 Mb) of the contig sequences anchored to 19 pseudo-chromosomes (Fig. 1; Supplementary Fig. 1; Table 1; Supplementary Table 5). The high quality, continuity, and completeness of the assembled genome were supported by a high mapping rate (99.4%) of Illumina short reads and 97.8% of the single-copy orthologs from the Benchmarking Universal Single-Copy Orthologs (BUSCO) analysis (Supplementary Table 6).

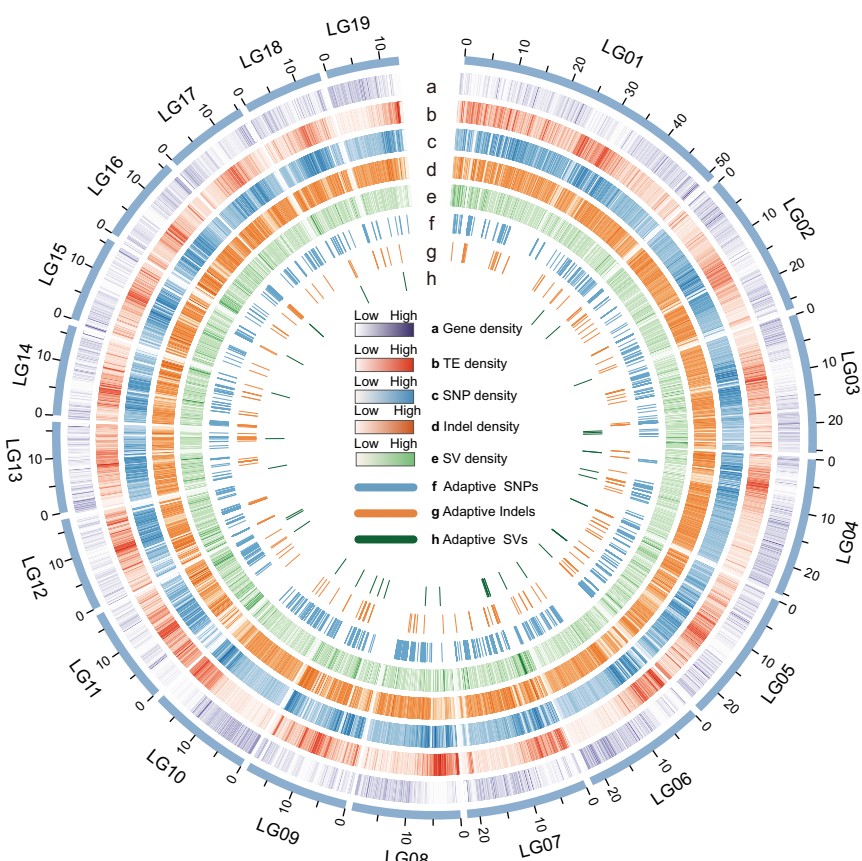

**Fig. 1 | Circos display of the genomic features and diversity of the assembled *P. koreana* genome.** Tracks from outermost to innermost are gene density (**a**), transposable element (TE) density (**b**), the distribution of SNPs (**c**), indels (**d**), and SVs (**e**); and the distribution of potentially climatic-adaptive SNPs (**f**), indels (**g**), and SVs (**h**) identified by LFMM across the genome. Source data are provided as a Source Data file.

**Table 1 | Statistics for the genome assembly and annotation**

| Genome assembly | |
|---|---|
| Assembled genome size (Mb) | 401.41 |
| Number of contigs | 135 |
| N50 of contigs (bp) | 6,410,956 |
| N90 contig length (bp) | 1,239,380 |
| Longest contig (bp) | 17,436,127 |
| Number of protein-coding genes | 37,072 |
| Percentage of repetitive sequence | 37.19 |
| GC content (%) | 35.12 |
| BUSCO (complete) (%) | 97.83 |

In total, 37.2% of the genome sequences were identified as repetitive elements with retrotransposons and DNA transposons occupying 16.0% and 17.9%, respectively (Supplementary Table 7). Through a combination of the transcriptome, homology, and ab initio-based approaches, we predicted 37,072 protein-coding genes with an average coding sequence length of 1136 base pairs (bp) and an average of five exons per gene (Supplementary Table 8). In addition, 95.4% of these genes (35,380) could be found in at least one public database (Pfam, InterPro, NR, Swiss-Prot, GO, and KEGG; Supplementary Table 9). We also identified a set of non-coding RNAs (Supplementary Table 10).

## Population structure, genetic diversity, and demographic history

We generated whole-genome resequencing data of 230 individuals from 24 populations sampled across the natural distribution of the species in Northeast China (Fig. 2a, b; Supplementary Data 1). On average, ~95% of the clean reads were aligned onto the reference genome, with an average depth of 27.4× and coverage of 94.6% (Supplementary Data 1). Using this dataset, we identified a total of 16,619,620 high-quality SNPs and 2,663,202 indels (shorter than or equal to 50 bp). We also identified a final set of 90,357 large SVs (>50 bp). The genome-wide distribution patterns of these genetic variations were found to be consistent (Fig. 1c–e), and the observed site frequency spectrums were also similar (Supplementary Fig. 2). We first used ADMIXTURE to investigate the genetic structure and revealed three clusters (Fig. 2a, b; Supplementary Fig. 3). Two clusters were in the southern region (Changbai Mountains area), and the third was in the northern (Greater Khingan Mountains area), but multiple individuals from both southern and northern groups showed an admixture of different clusters (Fig. 2a). Both principal component analysis (PCA) and neighbor-joining (NJ) clustering produced the consistent results (Supplementary Figs. 4a and 5a). We examined patterns of genetic differentiation and isolation-by-distance (IBD) between and within the geographical groups. We detected significant IBD in the southern group but not in the northern group, possibly owing to the small number of populations used (Supplementary Fig. 6a, c). As expected, the pattern of IBD was stronger for all populations combined than only populations from the southern or northern region alone (Fig. 2c). These results suggest that multiple refugia might have existed for this species during the Quaternary glacial periods, and the current distribution of populations is likely to have resulted from post-glacial re-colonization and secondary contact from different refugia[25]. However, we could not exclude the possibility that populations in the intermediate region connecting southern and northern groups might have been destroyed recently by human activities, which may have biased our hierarchical structure presented here[26].

We examined the extent of genetic divergence between southern and northern groups based on the following three analyses. First, we estimated the joint frequency spectrum of genetic variations and found that large proportions of variation were shared between the two geographical groups (Supplementary Fig. 7a). Genetic differentiation between them was estimated to be weak (Supplementary Fig. 7b; the average $F_{ST}$ values, 0.021). Similar levels of nucleotide diversity were revealed across the 24 populations, and the nucleotide divergence ($d_{xy}$) between the two groups was almost the same as the nucleotide diversity within groups (Supplementary Fig. 7c–e). Second, we used the pairwise sequentially Markovian coalescent (PSMC) to assess changes in effective population size ($N_e$) in the history (Supplementary Fig. 5b). We found that the inferred $N_e$ differed only between the southern and northern groups after the last glacial maximum (LGM, 10,000–20,000 years ago). The northern group showed a steady population decline, whereas a slight population expansion was observed in samples from the southern group. These inferred demographic histories were also confirmed by Tajima's $D$ statistics (Supplementary Fig. 5c) where a more negative Tajima's $D$ value was found for the southern group than for the northern group. Finally, the genome-wide decay of linkage disequilibrium (LD) as a function of physical distance showed similar patterns in the southern and northern populations, with $r^2$ declining below 0.2 after ~15 kbp on average (Supplementary Fig. 5d). Taken together, our results reveal relatively weak genetic differentiation between southern and northern groups and suggest that they may have diverged recently after the LGM.

## Identification of genomic variants associated with local climate adaptation

The high-quality reference genome for *P. koreana* coupled with the high-depth-resequencing data generated in this study facilitated the precise characterization of genomic information, including not only SNPs but also indels and SVs that are usually ignored[27]. We used two complementary genotype–environment association (GEA) approaches to detect the environment-associated genetic variants. First, we tested for GEAs for 19 environmental variables (10 temperature and nine precipitation-related variables; Supplementary Table 11) using the latent factor mixed model (LFMM)[28], which tests for associations between genotypes and environment variables while accounting for background population structure. We identified a total of 3013 SNPs, 378 indels, and 44 SVs (Fig. 1f–h), involving 514 genes that were significantly associated with one or more environmental variables (Fig. 3; Supplementary Fig. 8; Supplementary Data 2). These environment-associated variants were widely distributed across the genome and did not cluster in specific regions.

LFMM is a univariate approach that tests for associations between one variant and one environmental variable at a time, and, to alleviate these issues, we also used a complementary multivariate landscape genomic method, redundancy analysis (RDA)[29], to identify co-varying variants that were likely associated with multivariate environment predictors. After considering the ranked importance based on gradient forest analysis and the correlations among these environmental variables (Supplementary Fig. 9a), six variables with Spearman correlation coefficient |$r$| <0.6 were retained for the RDA analyses to avoid issues due to multicollinearity, including three temperature variables (annual mean temperature (BIO1), isothermality (BIO3), and maximum temperature of warmest month (BIO5)) and three precipitation variables (precipitation of wettest month (BIO13), precipitation seasonality (BIO15), and precipitation of coldest quarter (BIO19)). By visualizing climate-associated genetic variation across the natural distribution, we found that adaptive genetic variation could be largely explained by these six climatic variables (Supplementary Fig. 9b). Of the 3435 significant variants identified by LFMM, 1779 (1554 SNPs, 206 indels, and 19 SVs) were found to display extreme loadings (standard deviation >3) along one or multiple RDA axes (details in the "Methods" section). These shared variants were regarded as "core adaptive variants" for

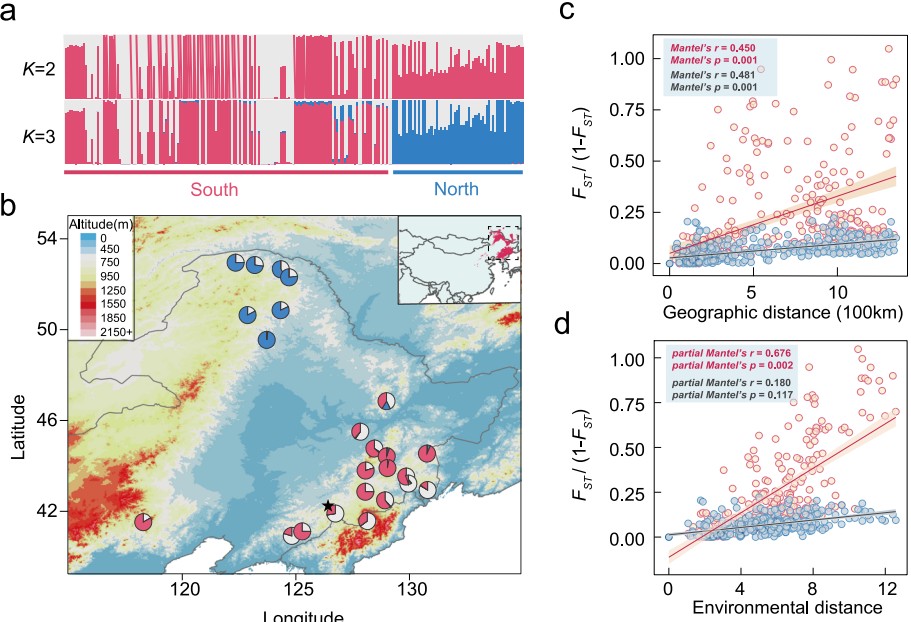

**Fig. 2 | Population genomic analyses of *P. koreana*. a** Model-based population assignment using ADMIXTURE with *K* from 2 to 3. The height of each colored segment represents the proportion of the individual's genome derived from inferred ancestral lineages. **b** Geographic distribution of 24 natural populations (circles) where colors represent ancestral components inferred by ADMIXTURE (according to the substructure at *K* = 3). The location of the individual selected for genome assembly is indicated by a black star. Inset: the current geographic range of *P. koreana* predicted by ecological niche models (ENMs). **c** Isolation-by-distance analyses (Mantel test, two-sided) for populations (*n* = 276) based on neutral (blue dots and black line) and adaptive variants (red dots and red line) separately. The shadow of linear regression denotes the 95% confidence interval. **d** Isolation-by-environment analyses (partial Mantel test, two-sided, controlling for the effect of geographic distance) for populations (*n* = 276) based on neutral (blue dots and black line) and adaptive variants (red dots and red line) separately. The shadow of linear regression denotes the 95% confidence interval. Source data are provided as a Source Data file.

local climate adaptation, and more adaptive variants were observed to be associated with precipitation-related compared to temperature-related variables (Supplementary Fig. 10).

To evaluate the potential adaptive variants identified here, we first estimated and compared inter-population genetic differentiation ($F_{ST}$) between the climate-associated adaptive variants and the randomly chosen ones. Significantly stronger $F_{ST}$ values were observed at these adaptive variants (Supplementary Fig. 11), indicating that spatially varying selection likely plays an important role in driving genomic differentiation between populations[9,30]. In addition, we used Mantel and partial Mantel tests to assess patterns of IBD and isolation-by-environment (IBE) for the potentially adaptive and neutral variants, respectively (Fig. 2c, d; Supplementary Fig. 6). We found that both adaptive and neutral variants displayed significant patterns of IBD within and between geographic groups, although adaptive variants showed slightly stronger pattern compared to neutral variants (Fig. 2c; Supplementary Fig. 6a, c). However, in contrast to the weak pattern of IBE observed for the neutral variants after controlling for the effect of geography, adaptive variants showed a strong and significant IBE in partial Mantel tests (Fig. 2d; Supplementary Fig. 6b, d), suggesting that genetic variation of the adaptive variants was mainly influenced by the environment. Furthermore, adaptive structuring exhibited a different pattern from neutral genetic clustering, showing little association with geography and/or population structure (Supplementary Fig. 4a, b). To decompose the relative contributions of climate, geography, and population structure in explaining adaptive and neutral genetic variation, we performed partial RDA and found that the exclusive contribution of climate effects explained 41% of the genetic variation of adaptive variants, which was much higher than 10% of neutral variants when controlling for geography and population structure (Supplementary Fig. 4c, d). Overall, all these results suggest that the identified adaptive variants in our study should be relatively robust to the confounding effects of population structure and geographical factors and

were largely shaped by the environmental gradients across the landscape[29].

Of the core adaptive variants detected by both LFMM and RDA, only 3.2% were non-synonymous, and 2.0% were synonymous mutations, with all remaining variants being non-coding (Supplementary Table 12), indicating that adaptation to climate has primarily evolved as a result of selection acting on regulatory rather than on protein-coding changes[31]. In particular, these climate-adaptive variants were enriched to be located in the 5' untranslated region (UTR) of genes and transposable elements (TEs) (Supplementary Fig. 12). Approximately 9.7% of these variants were located within the regions of accessible chromatin as identified by transposase-accessible chromatin sequencing (ATAC-seq) (Supplementary Data 2), again suggesting that mutations in the cis-regulatory elements may play important roles in driving environmental adaptation in natural populations of this species. To further assess the selection pressures acting on the climate adaptive variants, we calculated the standardized integrated haplotype score (iHS) across all common variants to identify loci with signatures of selective sweeps[32]. Our results show that climate-associated variants did not display stronger signatures of positive selection compared to randomly selected SNPs (Supplementary Fig. 13), suggesting that adaptation to the local climate in *P. koreana* may largely arise by the polygenic selection, characterized by subtle to moderate shifts in allele frequencies of many loci with small effect sizes[33,34].

## Geographic distribution of the variants in the genes with local adaptation

Many genes previously reported to be involved in climate adaptation were identified here with variants to be associated with environmental variables (Supplementary Fig. 8; Supplementary Data 2; Supplementary Table 13), although no significant functional enrichment could be detected. The variants of these genes related to precipitation showed similar geographic distribution in frequencies (Supplementary Fig. 14).

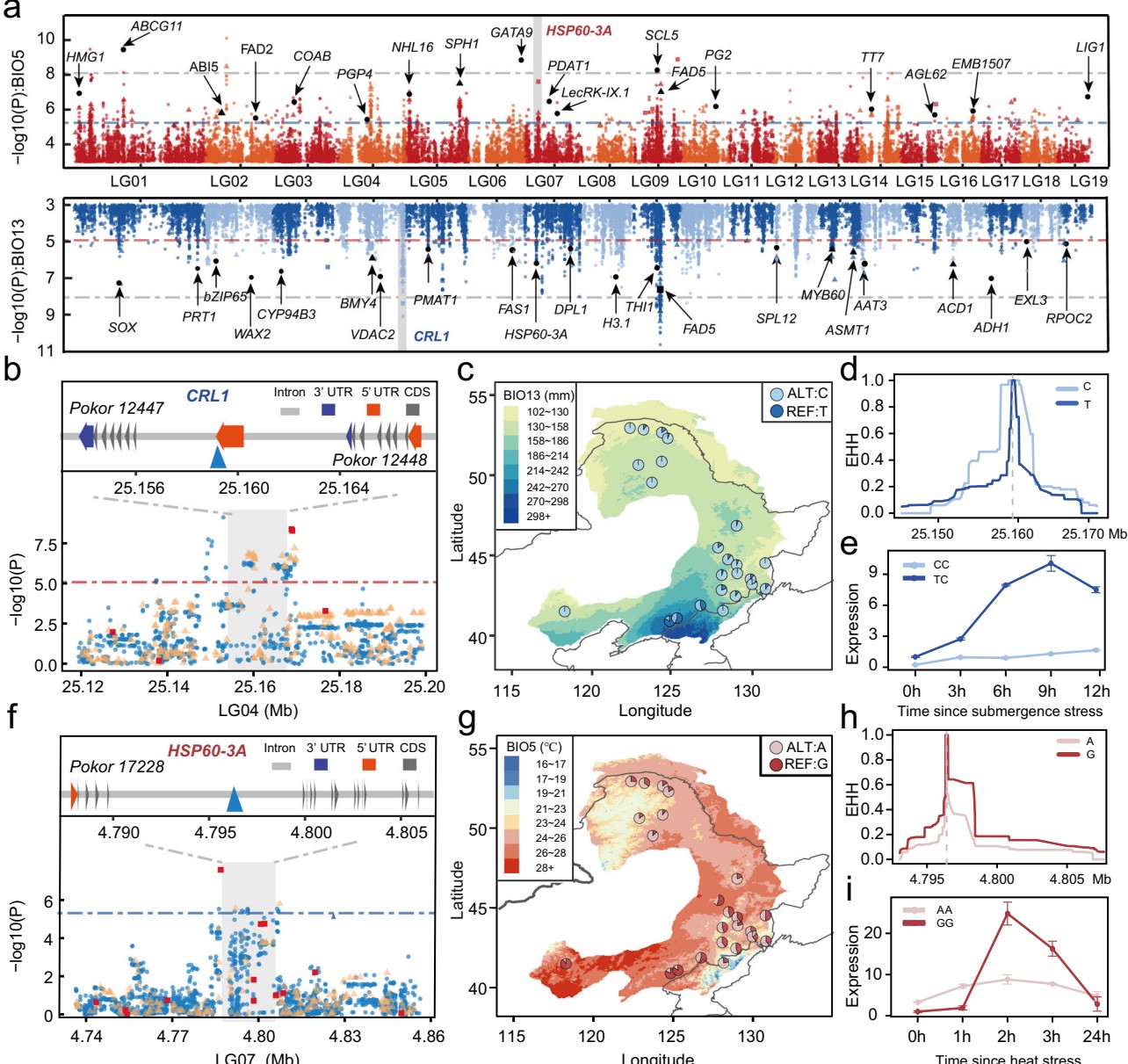

**Fig. 3 | Genome-wide screening of the loci associated with local environmental adaptation. a** Manhattan plot for variants associated with the Maximum Temperature of the Warmest Month (BIO5) (red, upper panel) and the Precipitation of the Wettest Month (BIO13) (blue, lower panel). Dashed horizontal lines represent significance thresholds (blue or red represents the FDR correction, adjusted $P = 0.05$; gray represents the Bonferroni correction, adjusted $P = 0.05$). Selected candidate genes are labeled in the plot at their respective genomic positions. **b, f** Upper panels: the gene structure of *CRL1* (**b**) and *HSP60-3A* (**f**) (blue triangles: representative candidate adaptive SNPs corresponding to the sites shown in **c–e** and **g–i**). Lower panels: local magnification of the Manhattan plots (blue circles:

SNPs; yellow triangles: indels; red squares: SVs) around the selected genes (gray shadows). **c, g** Allele frequencies of candidate adaptive SNPs (**c**, LG04:25159299; **g** LG07:4796402) associated with BIO5 (**c**) and BIO13 (**g**) across the 24 populations. Colors on the map are based on variations of the relevant climate variables across the distribution range. **d, h** Decay of EHH for two alternative alleles around LG04:25159299 (**d**) and LG07:4796402 (**h**). **e, i** Dynamic relative expression level of *CRL1* (**e**) and *HSP60-3A* (**i**) genes between the two genotypes using qRT-PCR under submergence (**e**) and heat (**i**) treatments. Error bars represent standard deviation, $n = 3$ biologically independent samples. Source data are provided as a Source Data file.

For example, the gene *CRL1* variants are strongly associated with precipitations during the wettest month (Fig. 3a). This transcription factor with a lateral organ boundary (LOB) domain plays an essential role in root development in response to flooding and drought stresses[35,36]. We found two tandem *CRL1* duplicates (*Pokor12247* and *Pokor12248*) (Fig. 3b) and identified a total of 104 candidate adaptive variants (83 SNPs, 19 indels, and two SVs). We chose one candidate adaptive SNP located in the 5' UTR of *Pokor12247* (LG04:25159299) as an example to show the distribution pattern of allele frequencies (Fig. 3b, c). The T allele was mainly distributed in the southeast region

that is characterized by heavy precipitation in the wettest month, whereas the C allele was almost fixed in areas experiencing low rainfall (Fig. 3c). To verify functional differentiation of the *Pokor12247* alleles in mediating adaptation to extreme precipitation, we performed qRT-PCR to profile its expression under submergence stress. The individuals with the TC genotype displayed enhanced expression compared to those CC individuals in response to submergence (Fig. 3e). This indicates that the individuals carrying the T allele may be associated with increased tolerance to submergence. Nevertheless, the relatively high degree of LD (Supplementary Fig. 15a) in this region makes it

difficult to identify the true causal mutation(s) that are involved in mediating environmental adaptation. Furthermore, we did not observe signals of strong recent selection at this locus[37]. The extended haplotype homozygosity (EHH) did not exhibit significant differences between haplotypes carrying the T or the C allele at the focal SNP (Fig. 3d; standardized |iHS| score = 1.693), which again supports a polygenic pattern of adaptation[38]. In addition, many other genes were also found to be involved in precipitation-associated adaptation (Supplementary Figs. 8 and 14; Supplementary Data 2; Supplementary Table 13), including *Pokor27800*, the ortholog of which is involved in regulating stomata-related stresses[39]; *Pokor18547*, the ortholog of which encodes a sphingoid long-chain base-1-phosphate lyase that is involved in the dehydration stress response[40]; and *Pokor25841*, the ortholog in *Arabidopsis* (SPL*12*) was found to regulate stress responses[41].

We also identified a set of temperature-associated loci, including genes orthologous to *Arabidopsis HMG1*, *PGP4*, *FAD5*, and *EMB1507*, showing similar allele frequency distribution patterns as we saw for the precipitation-associated genes (Fig. 3a; Supplementary Figs. 8 and 16; Supplementary Data 2). A striking example of such a gene associated with variation in the maximum temperature of the warmest month was *Pokor17228*, which encodes a heat shock protein (HSP) orthologous to *Arabidopsis HSP60-3A*[42]. The rapid synthesis of HSPs induced by heat stress can protect cells from heat damage and enable plants to obtain thermotolerance by stabilizing and helping refold heat-inactivated proteins[43]. Relatively high LD was found within the region surrounding this gene (Supplementary Fig. 15b), including a total of 62 candidate adaptive variants (59 SNPs, two indels, and one SV). We chose one candidate adaptive SNP located in an intronic region of *Pokor17228* (LG07: 4796402) to show the geographic distribution of the allele frequencies (Fig. 3f). Populations with a relatively high temperature of the warmest month of the year were more likely to carry the G allele, whereas the A allele was mainly observed in regions with low temperatures (Fig. 3g). We then examined expression patterns of this gene in response to heat stress and found that the GG genotypes showed much higher expression than the AA ones after 2 and 3 h of heat stress treatment (Fig. 3i), indicating that *Pokor17228* is a likely candidate gene for heat stress tolerance in *P. koreana*. We similarly failed to detect signatures of strong recent selection signals at this locus (standardized |iHS| score = 1.661). Despite this, the haplotypes carrying the warm-adapted allele (G) had elevated EHH relative to the haplotypes carrying the other allele (A) (Fig. 3h), suggesting it might have experienced weak positive selection. Our analyses of these selected genes support a polygenic model for local climate adaptation across natural populations of *P. koreana*. The thorough characterization of the genetic basis underlying ecological adaptation performed in this study offers promising information for predicting species' response to future climate change[15,20].

## Genomic offset prediction for future climate change

Based on the established contemporary genotype–environment relationships and the identified climate-associated genetic loci, we predict how populations of *P. koreana* will respond to future climate change. For the future climate projections, we integrated the prediction of the genomic offset across four various future climate models to account for the variability between models. In addition, we also considered two different emission scenarios of the shared socioeconomic pathway (SSP126 and SSP370) among the four adopted by the CMIP6 consortium for two defined periods (2061–2080 and 2081–2100)[44]. Three complementary approaches were used to investigate the spatial pattern of maladaptation across the range of *P. koreana* under future climate conditions.

We first calculated the risk of non-adaptedness (RONA), which measures the expected allele frequency shifts required to cope with future climate conditions after establishing a linear relationship

between allele frequencies at environmentally associated variants and present climates[18,45]. We found that although the values of RONA varied across the four future climate models (Supplementary Fig. 17; Supplementary Data 3), the estimates were highly correlated across populations (Supplementary Fig. 18). Therefore, the average RONA values across models were inferred, and compared between populations. As expected, for most environmental variables, RONA increases under more severe climate change scenarios, with higher emissions leading to increased overall RONA values (i.e., SSP370 versus SSP126; Fig. 4 and more details in Supplementary Data 3). There was substantial variation in RONA estimates among both environmental variables and populations (Fig. 4; Supplementary Figs. 19–21). We chose predictions for two environmental variables (BIO5 and BIO13, described above) as a representative outcome and found that populations located in areas with more drastic environmental changes were anticipated to have greater RONA values (Fig. 4a, c). In addition, we observed that the RONA estimates across the three types of variants (SNPs, indels, and SVs) were highly consistent (Supplementary Fig. 22). In contrast to the equally large values of RONA observed in both northern and southern distributions of *P. koreana* in face of temperature changes (Fig. 4a, b), the southeastern populations near the Korean Peninsula, which were predicted to experience severe rainfall and extreme precipitation events in the future, displayed much higher RONA values compared to others for precipitation-related variables (Fig. 4c, d; Supplementary Fig. 21).

We then used the gradient forest (GF) approach to model the turnover in allele frequencies along present environmental gradients and predict genetic offset to a projected future climate[17]. Compared to RONA which is estimated at the level of a single locus under a given environmental variable, the GF is an extension of the random forest approach that models the associations between the composite effects of many putatively adaptive locus to multi-climate variables simultaneously[46]. Consistent with RONA, the GF estimates of genetic offsets across the four future climate models were highly correlated (Supplementary Fig. 23), and the genetic offsets were measured as the averages across models. We further used all 19 climatic variables and the six uncorrelated variables that were the same as used in the RDA analyses to estimate the genetic offset across the geographic distribution of the species. The results showed similar spatial genomic offset patterns (Fig. 5a, b; Supplementary Fig. 24a, b; Supplementary Fig. 25), and therefore in the following, we present only the results using all 19 climatic variables. The comparison of genetic offset under different emission scenarios showed an increasing trend with rising emissions (Fig. 5a, b). Spatial mapping of the genomic offset similarly identified the southeastern populations near the Korean Peninsula as being most vulnerable to future climate change (Fig. 5a, b). The same results were obtained when using the independent datasets of SNPs, indels, and SVs (Supplementary Fig. 26).

In addition to quantifying the in situ maladaptation of populations as performed above, we further assessed the metrics of forward and reverse genetic offset that integrates migration into the analyses in addition to the classic (local) genetic offset[23]. With different maximum dispersal distances (100, 250, 500, 1000 km, and unlimited), we revealed largely consistent patterns (Supplementary Fig. 27), although restricting the maximum migration distances unavoidably resulted in higher forward offset (Supplementary Fig. 28). We, thus, estimated the forward genetic offset by identifying the minimum predicted offset, assuming that specific contemporary population can migrate to any location in the Eurasian continent. Furthermore, after shifting the focus from populations to locations, reverse genetic offset was calculated by identifying the minimum offset for any contemporary population in the current range that best matches the projected future climate of a specific location[23]. Although the predicted patterns of local, forward, and reverse offsets varied throughout the range of *P. koreana*, the southeastern populations near the Korean Peninsula

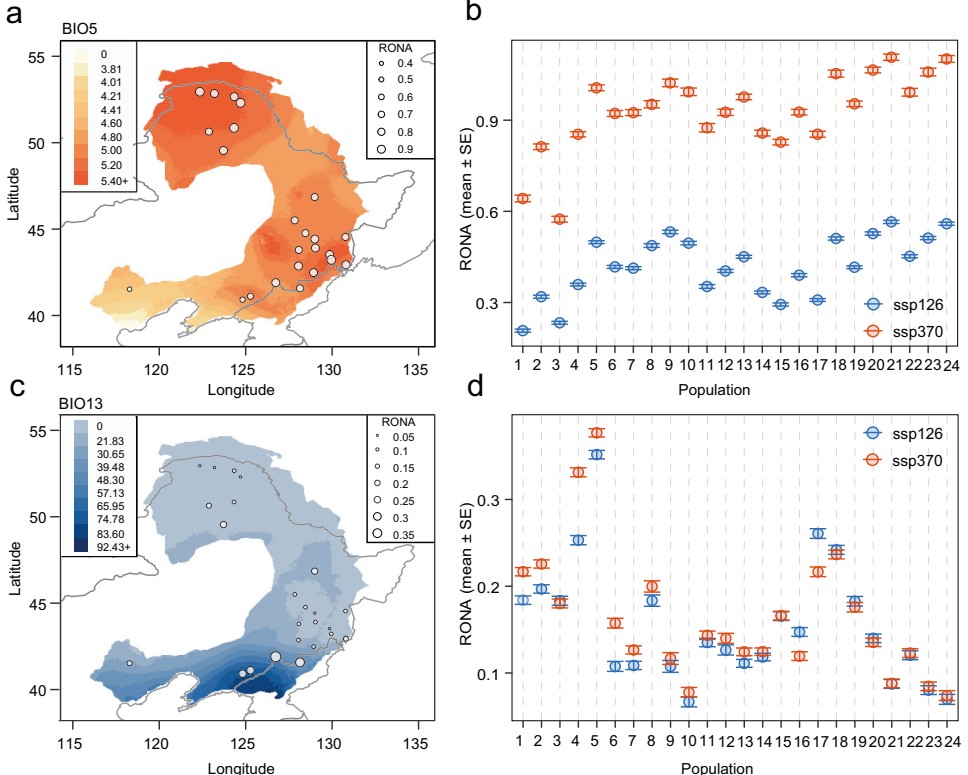

**Fig. 4 | RONA of *P. koreana* to future climatic conditions. a, c** The average RONA estimates across four climate models for the 24 populations under the SSP370 climate scenarios in 2061-2080 (**a** BIO5; **c** BIO13). The raster colors on the map represent the degree of projected future climate change (absolute change). Areas with darker red (**a**) or blue (**c**) are predicted to experience more substantial change in the respective climate variables. The size of the circles on the map represents the RONA values of different natural populations. **b, d** Comparison of the average RONA values under two different climate scenarios (SSP126 and SSP370) in 2061–2080 across populations for BIO5 (**b**) and BIO13 (**d**). Error bars represent the standard error (SE) of the average RONA calculated from four different climate models for BIO5 (*n* = 271 climate-associated variants) and BIO13 (*n* = 841 climate-associated variants), respectively. Source data are provided as a Source Data file.

were consistently predicted to have relatively high local, forward, and reverse offsets (Fig. 5c, d; Supplementary Fig. 24c, d). Therefore, our results indicate that no populations either locally or elsewhere in the range of this species pre-adapt to future climates in this region. Moreover, the present populations in this region cannot be mitigated by migration or dispersal to more suitable habitats[23]. Considering that these populations contain many unique, climate-adaptive genetic resources where a set of adaptive alleles for warm and wet climates have been identified in multiple functional important genes, more conservation and restoration efforts are highly necessary for populations in this area[47].

We finally examined whether the populations with higher genetic offset to future climate change also have an increased burden of deleterious mutations. We predicted and classified coding SNPs into four categories with respect to their effects using SIFT4G: synonymous, tolerated, deleterious, and loss of function (LOF)[48]. We used the ratios of derived functional (including tolerated, deleterious, or LOF variants) to synonymous variants as proxies for the genetic load. Together no relationship was observed between the predicted genetic offsets (including local, forward, and reverse offsets) and both genetic diversity and genetic load across populations, even for the LOF variants that are predicted to be strongly deleterious (Supplementary Figs. 29a–d and 30a–d). We also failed to reveal the associations between the SV burden and genetic offset (Supplementary Figs. 29e and 30e). As the analysis of genetic offset or the prediction of future climate maladaptation is based on putatively adaptive variations, whereas the measures of genetic load depend on the genome-wide distribution of deleterious mutations, it is not surprising to observe the minimal relationship between them[49].

## Discussion

Ongoing climate change is predicted to threaten populations of numerous species[1]. Despite the importance of intraspecific adaptive variations in mitigating such risks[15], predictions of range shift and population vulnerability to future climate change are still challenging without genome-scale knowledge. In this study, based on the well-assembled reference genome and population-level whole-genome resequencing data, we extracted all genetic variations to explore the genetic architecture of climatic adaptation in one key forest-dominant tree, *P. koreana*, in East Asia. We identified the genes and the corresponding variants that are correlated with local adaptation. We further revealed that the local adaptation of *P. koreana* to the current variable environments evolved by small polygenic allele frequency shifts as found for many other species[34,38]. After combining space-for-time and machine-learning approaches to predict the spatiotemporal responses of this species to future climate change, we identified a set of populations located in the southeastern part of the current distribution range as being most vulnerable under future climate scenarios. These populations are needed to be conserved with special management strategies not only because of their high genomic offset to future climate change but also because they contained many unique, climate-adaptive genetic resources[47].

However, it should be noted that all of the present genomic predictions of maladaptation to future climates must be used with caution, and further empirical validations are needed to confirm these findings. One of the promising approaches to validate these genomic predictions is linking the estimated genetic offset with the observed decrease in fitness of genotyped individuals from different populations through common garden experiments or controlled

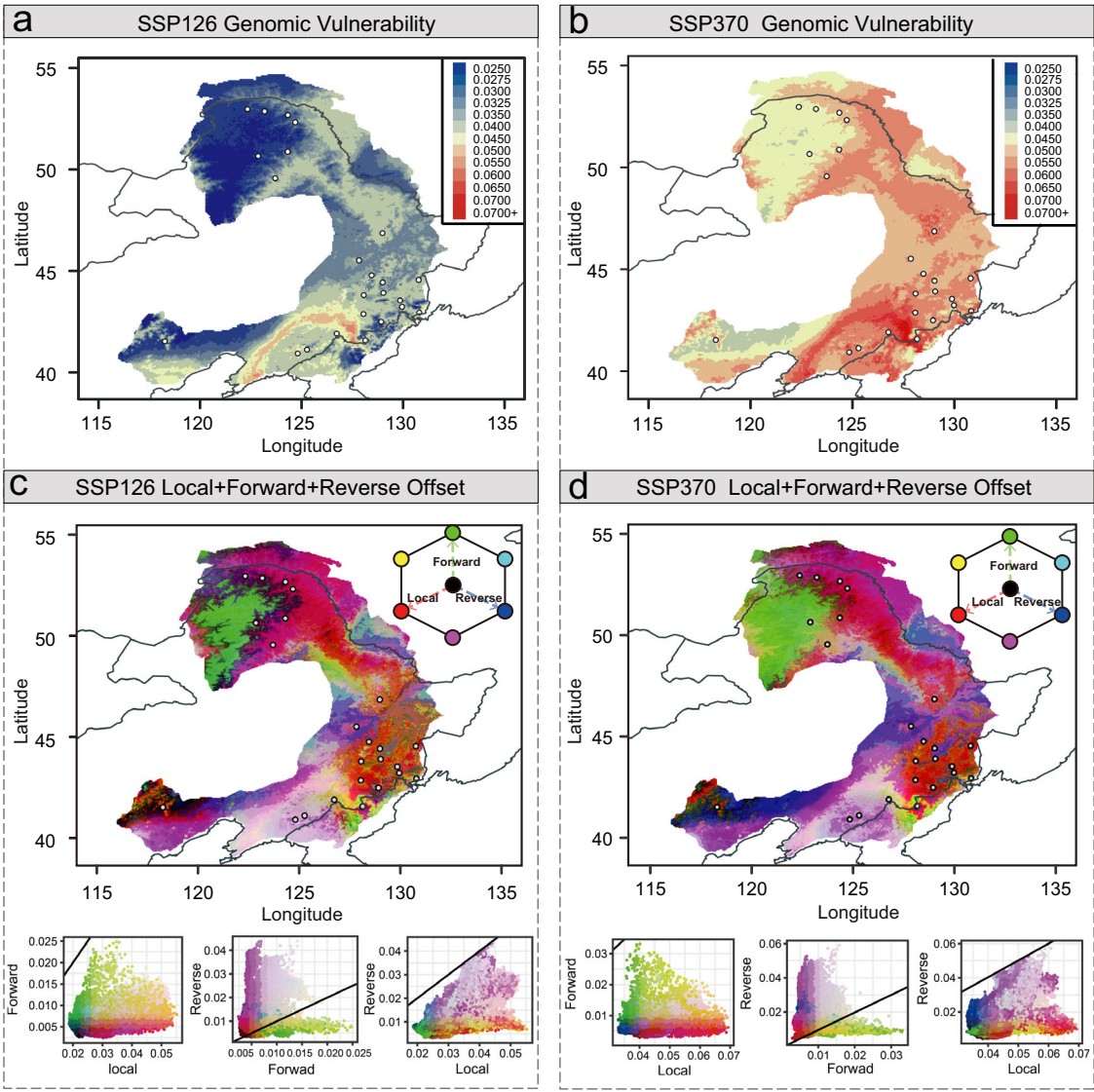

**Fig. 5 | Predicted genetic offsets to future climate change under SSP126 and SSP370 in 2061-2080. a, b** Map of the GF-predicted genetic offset averaged across four climate models across the distribution of *P. koreana* (*n* = 60,000 grids) under two scenarios of shared socioeconomic pathways SSP126 (**a**) and SSP370 scenarios (**b**) in 2061–2080. The color scale from blue to red refers to increasing values of genetic offset, and points on the map reflect the 24 sampled populations. **c, d** RGB map of local (red), forward (green), and reverse (blue) offsets throughout the range of *P. koreana* (*n* = 60,000 grids) under SSP126 (**c**) and SSP370 (**d**) scenarios in 2061-2080. Brighter cells (closer to white) have relatively high values along each of the three axes, whereas darker cells (closer to black) have relatively lower values. Lower panels are the bivariate scattergrams of **c** and **d** with 1:1 lines. Source data are provided as a Source Data file.

environment tests. In doing so, the fitness-relevant traits could be assessed by manipulating sampled populations to future climate conditions[11,50]. In addition, despite the high-quality genomic data facilitated us to explicitly include both SNPs, indels, and SVs in analyses and also revealed that the climate adaptation of *P. koreana* had a polygenic genetic basis, the genetic variants with small effects in more genes may have been unavoidably missed especially given the reduced sensitivity owing to the required multiple testing correction in genome scans for detecting signatures of climate adaptation. Furthermore, we have to be aware that, although we used both single-locus-based (RONA) and multi-locus-based (GF) approaches to assess the climate change vulnerability of *P. koreana*, the complexity of polygenic adaptation (i.e., pleiotropy and genetic redundancy), has not been considered in current assessments. Therefore, future efforts should focus on integrating quantitative genetics and systems biology approaches to better model the genomic complexity of polygenic adaptation and to further improve the prediction of the genomic offset to climate change[33,51]. Finally, the potential of populations to respond to future

climate change has no relationship with their genetic load within *P. koreana*, and a more thorough understanding of the association between genetic load and population vulnerability under climate change could benefit from more future studies that incorporate evolutionary processes into the prediction of species responses to climate change[52].

## Methods

### Plant materials and genome sequencing

Fresh leaves of a wild *P. koreana* plant in the Changbai Mountains of Jilin province in China were collected, and the total genomic DNA was extracted using the CTAB method. For the Illumina short-read sequencing, paired-end libraries with insert sizes of 350 bp were constructed and sequenced using an Illumina HiSeq X Ten platform. For the long-read sequencing, the genomic libraries with 20-kbp insertions were constructed and sequenced using the PromethION platform of Oxford Nanopore Technologies (ONT). For the Hi-C experiment, approximately 3 g of fresh young leaves of the same *P. koreana*

accession was ground to powder in liquid nitrogen. A sequencing library was then constructed by chromatin extraction and digestion, DNA ligation, purification, and fragmentation[53] and was subsequently sequenced on an Illumina HiSeq X Ten platform.

## Genome assembly and scaffolding

The quality-controlled reads were first corrected via a self-align method using the NextCorrect module in the software NextDenovo v2.0-beta.1 (https://github.com/Nextomics/NextDenovo) with parameters "reads_cutoff=1k (filter reads with length <1kbp) and seed_cutoff=32k (minimum seed length = 32kbp)". Smartdenovo v1.0.0 (https://github.com/ruanjue/smartdenovo) was then used to assemble the draft genome with the default parameters. To improve the accuracy of the draft assembly, two-step polishing strategies were applied. The first step included three rounds of polishing by Racon v1.3.1[54] based on the corrected ONT long reads. The second step included four rounds of polishing by Nextpolish v1.0.5[55] based on cleaned Illumina short reads after removing adapters and low-quality reads using fastp v0.20.0[56] with parameters '-f 5 -F 5 -t 5 -T 5 -n 0 -q 20 -u 20' (parameters '-f 5 −F 5 -t 5 −T 5' were used to trim five bases in the front and tail for both read1 and read2; parameters '-n 0 -q 20 −u 20' were used to keep reads with Phred quality >20, percent of unqualified bases <20, and with no N base). Finally, allelic haplotigs were removed using the purge_haplotigs v1.1.1[57] software with the options '-l 5 -m66 -h 170' (set read depth between 5 and 170 and the low point between the haploid and diploid peaks as 66) to obtain the final contig-level assembly.

For chromosome-level scaffolding, the Hi-C reads were first filtered by fastp v0.20.0 with the parameters described above. Each pair of the clean reads was then aligned onto the contig-level assembly by Bowtie2 v2.3.2[58] with parameters '-end-to-end, -very-sensitive -L 30'. The quality of Hi-C data was evaluated by HiC-Pro v2.11.4[59], which further classified read pairs as valid or invalid interaction pairs. Only valid interaction pairs were retained for further analysis. Finally, scaffolds were clustered, ordered and oriented onto chromosomes using LACHESIS[60] with parameters: CLUSTER MIN RE SITES = 100; CLUSTER MAX LINK DENSITY = 2.5; CLUSTER NONINFORMATIVE RATIO = 1.4; ORDER MIN N RES IN TRUNK = 60; ORDER MIN N RES IN SHREDS = 60. The placement and orientation errors that exhibit obvious discrete chromosome interaction patterns were then manually adjusted.

The completeness of the genome assembly was assessed both by the representation of Illumina whole-genome sequencing short reads from mapping back read to the assembly using bwa v0.7.12[61] and by Benchmarking Universal Single-Copy Orthologs (BUSCO) v4.0.5[62] with the searching database of "embryophyte_odb10".

## Repeat and gene annotation

For repeat annotation, we used the Extensive de-novo TE Annotator (EDTA v1.9.3)[63], which incorporates well-performed structure-based and homology-based programs (including LTRharvest, LTR_FINDER, LTR_retriever, TIR-learner, HelitronScanner, and RepeatModeler) and subsequent filtering scripts, for a comprehensive repeat detection. Subsequently, TEsorter (v1.2.5, https://github.com/zhangrengang/TEsorter/)[64] was used to reclassify those TEs that were annotated as "LTR/unknown" by EDTA.

For gene annotation, we first used RepeatMasker v4.1.0[65] to mask the whole-genome sequences with the TE library constructed using EDTA. An integrated strategy that combined homology-based prediction, transcriptome-based prediction, and ab initio prediction was used to predict the protein-coding genes. For homology-based gene prediction, published protein sequences of six plant species, including *Populus euphratica, Salix brachista, Salix purpurea, Populus trichocarpa, Arabidopsis thaliana,* and *Vitis vinifera,* were downloaded and aligned onto the repat-masked genome by using the TBLASTN (ncbi-BLAST v2.2.28[66]) program with *E*-value cutoff setting of 1e⁻⁵, and GeneWise v2.4.1[67] was then used to predict gene models with default

settings. For transcriptome-based gene prediction, trimmed RNA sequencing reads from leaf, stem, and bud tissues were mapped to the reference genome using HISAT v2.2.1[68] with parameters "-max-intron-len (maximum intron length) 20000 −dta (report alignments tailored for transcript assemblers including StringTie) -score-min (set a function governing the minimum alignment score needed for an alignment to be considered "valid") L, 0.0, -0.4", and Trinity v2.8.4[69] was used for transcripts assembly with default parameters. Assembled transcripts were subsequently aligned to the corresponding genome to predict gene structure using PASA v2.4.1[70]. For the ab initio prediction, Augustus v3.3.2[71] was employed using default parameters after incorporating the transcriptome-based and homology-based evidence for gene model training. Finally, all predictions of gene models generated from these approaches were integrated into the final consensus gene set using EvidenceModelerv1.1.1[70]. After prediction, PASA was again used to update alternatively spliced isoforms to gene models and to produce a final gff3 file with three rounds of iteration.

In addition, we also performed non-coding RNAs (ncRNAs) annotation. Transfer RNAs (tRNAs) were identified using tRNAscan-SE v2.0.7[72] with default parameters. Ribosomal RNAs (rRNAs) were identified by aligning rRNA genes of *P. trichocarpa* v3.1 to the assembly using blast. The other three types of ncRNA (microRNA, small nuclear RNA, and small nucleolar RNA) were identified using Infernal v1.1.4[73] by searching the Rfam database v12.0[74].

For functional annotation, our predicted protein-coding genes were aligned to multiple public databases including NR, Swiss-Prot, TrEMBL[75], COG, and KOG using NCBI BLAST + v.2.2.31 with an *E*-value of 1e−5 as the cutoff[66]. Motifs and domains were annotated by searching against InterProScan (release 5.32-71.0)[76]. Gene ontology (GO) terms and KEGG pathways of predicted sequences were assigned by InterProScan and KEGG Automatic Annotation Server, respectively[77].

## Genome resequencing, read mapping, and variant calling

A total of 230 individuals were collected from 24 natural populations across the total distribution of the species. Within each population, individuals were sampled after ensuring that sampled individuals were at least 100 m apart from each other. Genomic DNA was extracted from leaves with a Qiagen DNeasy plant kit, and whole-genome paired-end sequencing was generated using the Illumina NovaSeq 6000 platform with a target coverage of 20× per individual.

For raw resequencing reads, we used Trimmomatic v0.36[78] to remove adapters and cut off bases from either the start or the end of reads if the base quality was <20. Trimmed reads shorter than 36 bases were further discarded. After quality control, all high-quality reads were mapped to our de novo assembled *P. koreana* genome using the BWA-MEM algorithm of bwa v.0.7.17[61] with default parameters. The alignment results were then processed by sorting and PCR duplicate marking using SAMtools v.1.9[79] and Picard v.2.18.11 (http://broadinstitute.github.io/picard/). For genetic variant identification, SNP and indel calling were performed using the Genome Analysis Toolkit (GATK v.4.0.5.1)[80] and its subcomponents HaplotypeCaller, CombineGVCFs, and GenotypeGVCFs to form a merged VCF file with "all sites" (including non-variant sites) included using the 'EMIT_ALL_SITES' flag. SV calling was performed using the software DELLY v0.8.3[81] with default parameters. We further performed multiple filtering steps to only retain high-quality variants for downstream analysis. For SNPs, SNPs with multi-alleles (>2) and those located at or within 5 bp from any indels were removed. In addition, after treating genotypes with read depth (DP) < 5 and genotype quality (GQ) < 10 as missing, SNPs with missing rate higher than 20% were filtered; for indels, those with muti-alleles (>2) and with QualByDepth (QD) < 2.0, strand bias estimated using Fisher's exact test (FS) > 200.0, StrandOddsRatio (SOR) > 10.0, MappingQualityRankSumTest (MQRankSum) <-12.5, ReadPosRankSum < -8.0 were removed. Indels with

missing rate >20% after treating genotype with DP < 5 and GQ < 10 as missing were further filtered out; for SVs, those with length <50 bp and with imprecise breakpoints (flag IMPRECISE) were removed. After treating genotypes with GQ < 10 as missing, we further filtered SVs with a missing rate >20%. Finally, we implemented the software SNPable (http://lh3lh3.users.sourceforge.net/snpable.shtml) to mask genomic regions where reads were not uniquely mapped, and we filtered out variants located in these regions. After these filtering steps, 16,619,620 SNPs, 2,663,202 indels, and 90,357 SVs were retained for subsequent analyses. The filtered variants were further phased and imputed using Beagle v4.1[82] and the effects of individual variants were annotated using SnpEff v.4.3[83] with "-ud 2000" to define the length of upstream and downstream regions around genes, with other parameters being set to default.

## Ecological niche modeling

To investigate the current (1970–2000) potential distribution range of *P. koreana* around China, we performed ecological niche modeling (ENM) using Maxent v.3.3.3[84] with 19 current bioclimatic variables (Supplementary Table 11). In addition to the geographic distribution data of our 24 natural populations (Supplementary Data 1), we also added another seven geographical data from the Chinese Virtual Herbarium (https://www.cvh.ac.cn/) and the Global Biodiversity Information Facility (https://www.gbif.org/zh/) into the ENM analyses, which was performed with default setting after excluding the highly correlated environmental variables (Spearman correlation coefficient >0.7).

## Population structure analysis

We first used PLINK v1.90[85] with the parameters "indep-pairwise 50 10 0.2" to extract an LD-pruned SNP set with minor allele frequency (MAF) > 5%, which yielded 535,191 independent SNPs to be used in the population structure analysis. First, we used ADMIXTURE v.1.3.0[86] with default parameters to investigate population genetic structure across all individuals, with the number of clusters ($K$) being set from 1 to 8. We then used the *rda* function from the R package *vegan* 2.6-2[87] to perform the PCA on the pruned SNPs. To further assess the relatedness between individuals, the identify-by-state (IBS) genetic distance matrix was calculated using the "-distance 1-ibs" parameter in PLINK v1.90. We constructed the NJ phylogenetic tree based on the distance matrix using MEGAX[88] and displayed the tree using FigTree v.1.4.4. Finally, for the IBD analysis, we first used VCFtools v0.1.15[89] to calculate the population differentiation coefficient ($F_{ST}$). The matrix of $F_{ST}$ ($1-F_{ST}$) and the matrix of geographic distance (km) among different groups of populations were then used for performing the Mantel tests using the R package *vegan*[87], with the significance being determined based on 999 permutations.

## Genetic diversity, linkage disequilibrium and demographic history analysis

To estimate and compare genetic diversity across populations of *P. koreana*, we calculated both intra-population ($\pi$) and inter-population ($d_{xy}$) nucleotide diversity after taking into account both polymorphic and monomorphic sites using the program pixy v0.95.0[90] over 100-kbp non-overlapping windows. In addition, Tajima's $D$ statistics were calculated using VCFtools v0.1.15 in 100-kbp non-overlapping windows for the northern and southern groups of populations using the complete SNP dataset, respectively. To further estimate and compare the pattern of LD among different groups of populations, PopLDdecay v.3.40[91] was used to calculate the squared correlation coefficient ($r^2$) between pairwise SNPs with MAF > 0.1 in a 100-kbp window and then averaged across the whole genome.

PSMC[92] was used to infer historical changes in the effective population size ($N_e$) of *P. koreana* using default parameters with the entire genomic dataset. We selected seven individuals from both

the northern and southern groups of populations to run the PSMC analyses, and 100 bootstrap estimates were performed per individual. Assuming a generation time of 15 years and a mutation rate of $3.75 \times 10^{-8}$ mutations per generation[93], we converted the scaled population parameters into $N_e$ and years.

## Identification of environment-associated genetic variants

We used two different approaches to identify environment-associated variants (SNPs, indels, and SVs) across the whole genome. We kept only common variants with MAF > 10%, including a total of 5,182,474 SNPs, 736,051 indels and 30,934 SVs, for these analyses. First, we used a univariate latent-factor linear mixed model (LFMM) implemented in the R package LEA v3.3.2[94] to search for associations between allele frequencies and the 19 BIOCLIM environmental variables[95]. Based on the number of ancestry clusters inferred with ADMIXTURE v.1.3.0, we ran LFMM with three latent factors to account for population structure in the genotype data. For each environmental variable, we ran five independent MCMC runs using 5000 iterations as burn-in followed by 10,000 iterations. $P$ values from all five runs were then averaged for each variant and adjusted for multiple tests using a false discovery rate (FDR) correction of 5% as the significance cutoff.

Second, we performed a redundancy analysis (RDA) to identify genetic variants showing an especially strong relationship with multivariate environmental axes[29,96]. RDA has been shown to be one of the best-performing multivariant GEA approaches and exhibits low false-positive rates[29]. After considering the ranked importance of the 19 environmental variables estimated using GF analyses with R package "gradientForest"[46] and correlations among the variables, six environmental variables (BIO1, BIO3, BIO5, BIO13, BIO15, and BIO19) with pairwise correlation coefficients $|r| < 0.6$ were selected for the RDA analyses using the R package *vegan*. Significant environment-associated variants were defined as those having loadings in the tails of the distribution using a standard deviation cutoff of 3 along one or more RDA axes.

To investigate and compare the role of geography and environment in shaping spatial genetic variation of adaptive (the 1779 adaptive variants identified by both LFMM and RDA) and neutral (the 535,191 LD-pruned SNPs as used for population structure analyses) variants, Mantel and partial Mantel tests were separately used to test for associations between $F_{ST}(F_{ST}/1-F_{ST})$ and geographic (IBD) and environmental (IBE) distance (after accounting for the geographic distance) with significance determined using 999 permutations in the R package *vegan*[87], where environmental distance was represented by Euclidean distance of all scaled environmental variables. In addition, we used partial RDA to quantify the relative contribution of geography, population structure, and environment in explaining the proportion of adaptive and neutral genetic variation. Three datasets were used: (1) six uncorrected environmental variables used as in the above RDA analysis ('clim'); (2) three proxies of population structure (population scores along the first three axes of PCA, 'struct'); and (3) population coordinates (latitude and longitude, 'geog') to characterize explanatory variables of climate, population structure, and geography. For the two RDA models, population allele frequencies of adaptive and neutral variants were used as the response variables, respectively. The significance of explanatory variables was assessed using 999 permutations with the function *anova.cca* of the R package *vegan*.

To further assess selection pressures acting on climate adaptive variants, we assessed the extended haplotype homozygosity (EHH) pattern for a selected set of strongly associated variants using the R package "rehh"[97] and calculated the standardized iHS across the genome for common variants using the software selscan v.1.3.0[98].

## Stress treatment and expression analysis by qRT-PCR

Stem segments from wild genotypes of *P. koreana* were surface sterilized by soaking in 10% sodium hypochlorite solution and 70% ethyl

alcohol for 5 min and then thoroughly washed five times with distilled water. The stem segments were inserted into MS medium (0.05 mg/L NAA) for 30 days at 25/20 °C (day 16 h/night 8 h), and, after rooting, the stem segments were transplanted to the soil for 40 days at 25/20 °C (day 16 h/night 8 h). To explore the effect of different genotypes of one candidate adaptive SNP located in the 5′ UTR of *Pokor12247* (LG04:25159299) in mediating adaptation to extreme precipitation, we carried out a submergence treatment. For the submergence treatment, water was maintained at 2 cm above the soil surface and plants were maintained in the growth chamber providing 25 °C/20 °C (day 16 h/night 8 h) for 0, 3, 6, 9, and 12 h. In addition, we also carried out a heat stress treatment to explore the effect of one candidate adaptive SNP located in the intronic region of *Pokor17228* (LG07: 4796402) in response to heat stress. For the heat stress treatment, plants were placed into a plant incubator at 42 °C/20 °C (day/night) with the illumination of 16/8 h (day/night) for 0, 1, 2, 3, and 24 h. At each time point, leaf tissues were collected from each plant at the same place and frozen immediately in liquid nitrogen for expression analyses.

Quantitative reverse transcription PCR (qRT-PCR)[99] was used to investigate the expression levels of selected genes in the abiotic treatments (*Pokor12247* for submergence stress and *Pokor17228* for heat stress). Total RNA was extracted from pooled leaf materials using a Plant RNA extract kit (Biofit, Chengdu, China), and the HiScript II RT SuperMix for qPCR kit (+gDNA wiper) (Vazyme, Nanjing, China) was used to obtain cDNA. qPCR was performed with gene-specific primers (Supplementary Table 14) using the Taq Pro Universal SYBR qPCR Master Mix (Vazyme, Nanjing, China) reaction system on the CFX96 Real-Time detection system (Bio-Rad). Each experiment was performed with three technical replicates, and the *UBQ10* was used as the endogenous control for data analysis.

### ATAC-seq analysis

For the ATAC experiment, fresh leaf tissues were collected from the same individual used for the genome assembly of *P. koreana* and prepared according to the experimental protocol following ref. 100. In brief, approximately 500 mg of flash-frozen leaves were immediately chopped and processed for ATAC-seq, followed by library construction, and were then subjected to sequencing on the Illumina HiSeq X-Ten platform. The raw reads generated were first trimmed using Trimmomatic v.0.36[78] with a maximum of two seed mismatches, and the adapters were trimmed by NexteraPE. Then, the clean reads were aligned to the reference genome using Bowtie v.2.3.2[58] using the following parameters: 'bowtie2 -very-sensitive -N 1 -p 4 -X 2000 -q' (the number of mismatches allowed for seed comparison was set to 1, the threads were set to 4, and the longest insertion clip length was set to 2000). Aligned reads were sorted using SAMtools v.1.1.1[79]. The redundant reads from PCR amplification and reads that mapped to either chloroplast or mitochondria were removed using Picard v.2.18.11 (http://broadinstitute.github.io/picard/). Finally, only high-quality properly paired reads were retained for further analysis. ATAC-seq peak calling was done by MACS2[101] with the '-keep dup all' function.

### Genomic offset assessment

For each sampling location, we downloaded future (2061–2080 and 2081–2100) environmental data for the 19 BIOCLIM variables from the WorldClim CMIP6 dataset of four different climate models (BCC-CSM2-MR model, ACCESS-CM2 model, CanESM5 model, and GISS-E2-1-G model; resolution 2.5 arcmin)[95]. Each of the two future environmental datasets consists of two shared socioeconomic pathways (SSPs): SSP126 and SSP370. We used three different approaches to evaluate the genomic offset to future climate change.

First, we calculated the RONA[18], which quantifies the theoretical average change in allele frequency needed to cope with climate change, under projected future climate scenarios. Following the method used in ref. 24, a linear relationship between allele frequencies

at significantly associated loci (detected by both LFMM and RDA) and environmental variables was first established using linear regressions. For each locus, population, and environmental variable, the theoretical allele frequency change needed to cope with future climate conditions (RONA) was calculated for each of the four climate models and then combined, and the average RONA values were further weighted by the $R^2$ for each linear regression following ref. 45. In addition, to explore and compare the patterns of RONA calculated by different types of adaptive variants (SNPs, indels, and SVs), we further calculated RONA using the three separated datasets, respectively, for two representative environmental variables (BIO5 and BIO13).

Second, as a complementary approach to RONA, we used a non-parametric, machine-learning GF analysis to calculate genomic offset across the range of *P. koreana* using 'gradientForest' in R[17,46]. For each given climate model, we built a GF model for estimating the genetic offset under the different future scenarios with both the 19 environmental variables and the six environmental variables. The genetic offset was calculated as a metric for the Euclidean distance of the genomic composition between the current and future projected climates and then mapped with ArcGIS 10.2 to display its geographical distribution. Given the high correlation of genetic offset that was estimated across models, the average values of genetic offset across models were used to predict the local maladaptation to future climate. Same as RONA, the genetic offset was also calculated and compared among the three different types of variants (SNPs, indels, and SVs).

Third, following the approach used in ref. 23, we integrated migration to predict potential maladaptation to future climate change and calculated three different formulations of genetic offset: the local, forward, and reverse offsets for each climate model. After quantifying the correlations of local, forward, and reverse offsets across the four different climate models, we predicted the three genetic offset metrics by calculating the average values across the climate models. For forward genetic offset, we further assessed its sensitivity to dispersal constraints and tested how forward offset varied when the maximum allowable migration was limited to different distance classes, including 100, 250, 500, and 1000 km, and unlimited to any location in the Eurasian continent. Furthermore, to visualize local, forward, and reverse offsets simultaneously, we mapped these three metrics as the red, green, and blue bands of an RGB image, respectively, as in ref. 23.

Finally, we explored whether there was an association between the genomic offset to future climate change and the accumulation of genetic load across populations of *P. koreana*. To assess the deleterious genetic load carried by each population, the effects of SNP variants on protein-coding gene sequences were first annotated and categorized as LOF, deleterious (SIFT score < 0.05), tolerated (SIFT score < 0.05), or synonymous based on the sorting intolerant from tolerant (SIFT) algorithm implemented in SIFT4G software using UniRef100 as the protein database[48]. The derived versus ancestral allelic state was determined at each SNP position using the est-sfs software[102] through comparison with *P. trichocarpa* sequences[103]. Then, the ratio between the number of derived mutations at LOF, deleterious, and tolerated sites relative to the number of synonymous variants was calculated and used as proxies for genetic load per population. In addition, as SVs are, on average, deleterious, we further calculated the SV burden represented by the averaged ratio of heterozygous SV to heterozygous SNP across individuals for each population. Finally, we used the *cor.test* function in R to calculate the Spearman's correlation coefficients between the three metrics of genetic offsets (local, forward, and reverse) under two future scenarios (SSP126 and SSP370) in 2061–2080 and the above proxies of genetic load across the 24 populations, respectively.

### Reporting summary

Further information on research design is available in the Nature Research Reporting Summary linked to this article.

## Data availability

All data needed to evaluate the conclusions in the paper are present in the paper and/or the Supplementary Materials. All sequencing data, including the assembled genome, the raw data for genome assembly and annotation (Nanopore long reads, Illumina reads of whole-genome sequencing, transcriptomes, ATAC-Seq data and Hi-C reads), and whole-genome resequencing data for 230 individuals in this study have been deposited in the National Genomics Data Center (https://ngdc.cncb.ac.cn) under accession number PRJCA008692. Source data are provided with this paper.

## Code availability

All scripts used in this study are available at https://github.com/jingwanglab/Populus_genomic_prediction_climate_vulnerability.

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

## Acknowledgements

This work was supported by the National Key Research and Development Program of China (2022YFD2201200 to J.W., 2021YFD2200202 to J.L.), National Natural Science Foundation of China (31971567 to J.W.) and Fundamental Research Funds for the Central Universities (SCU2022D003, SCU2021D006, SCU2019D013 and 2020SCUNL207 to J.L., K.M., J.W.).

## Author contributions

J.W., K.M., and J.L. conceived the research. J.W. supervised the work. Y.S., H.Z., and K.M. performed the sampling and collected the materials. Y.S., Z.L., T.S., C.J. X.Z., Q.L., G.Y., and X.X. conducted all bioinformatics analyses. X.D., J.F., H.L., and Y.J. performed the experiments. Y.S., Z.L., and J.W. wrote the manuscript, with input from P.K.I and J.L. All authors approved the final manuscript.

## Competing interests

The authors declare no competing interests.
