## [Peer Review File · Nature Communications]

Genomic insights into local adaptation and future climate-induced vulnerability of a keystone forest tree in East AsiaReviewers' Comments:

Reviewer #1:

Remarks to the Author:

This manuscript represents an impressive amount of effort to use genomic data to test for climate adaptation and quantify populations' vulnerability to climatic change using. Using whole-genome sequences from 230 individuals sampled from across 24 populations the authors combine evaluation of population genomic statistics with environmental association analyses to identify SNPs, indels, and SVs potentially involved in adaptation to climate. They then use this understanding of the genetic basis of contemporary climate adaptation to evaluate genomic vulnerabilities using two metrics, risk of non-adaptedness and genomic offsets to identify regions within the species' range that may be more at risk to changing climatic conditions. Overall, I was extremely impressed with this work and the very careful approach the authors took to data analysis. In general, the manuscript is extremely well-written and clear (but not some parts of the introduction could benefit from clarification) and the methods and figures/tables easy to follow. My two main concerns stem from what might be a philosophical consideration, but should be justified in a revision. Specifically (i) understanding when to consider from an analytical and applied conservation perspective genetic clusters independently (as was done for some analyses – but not all) or as representing standing genetic variation for all clusters together across a species' range and (ii) given the degree to which the authors identify and advocate for the polygenic basis of climate adaptation how can data from individual genes (CRL1/HSP60-3A) provide us with respect to risk of non-adaptedness. I elaborate on both below and provide also some minor comments for revision.

One of the points I found a little confusing in the manuscript was the fact that Line 204 suggests overall there is weak population structure between southern and northern groups – but the admixture analysis supports a cluster of $K=3$ (Line 164) separating the northern and southern genetic clusters and sometimes they are analyzed together and sometimes apart. To address this, I think first that Figure 2 would really benefit from an understanding of the rangewide distribution of *Populus koreana* – it is unclear if there is potentially a sampling bias that might be contributing to the identification of genetic clusters and if sampling occurred in the northeastern areas that connect the two regions if a gradient in allele frequencies rather than distinct clusters would be observed. Figure 2C suggests there is likely a pattern of IBD suggesting patterns of post-glacial recolonization, but it was also interesting to note that the authors did not subset the genetic variation used to assess IBD (which could be assessed for neutral loci independently from adaptive loci). Given the lack of overlap in the climatic distribution of the northern and southern clusters (Figure S12) it's a little unclear if populations are exhibiting isolation by distance or potentially isolation by environment (or both). So from Figure S12(d) it seems that there is limited overlap in the environment for the genetic clusters (for BIO1, 2, 4, etc.) so for assessing linear regressions of allele frequencies with environmental variables – does it make sense to model the entire range or would it make sense to model within clusters (a regression for each of the northern and southern clusters). From an applied perspective and thinking how this data may inform conservation priorities and management this may provide a distinct perspective for the relationships of environment with the different clusters. Indeed, for the risk of non-adaptedness (RONA) you model the clusters independently, suggesting that the genetic variation underlying climate adaptation and adaptive capacity of clusters is different. It's also unclear what the black regression line is based on in Figure 4. Given these two distinctions, I think there is opportunity to (a) justify sampling and indicate the entire distribution of *P. koreana* and relate potential biases that may have occurred related to sampling (b) consider whether GEAs should be examined within each genetic cluster or across the species' range as a whole – where components of both approaches have been taken here. From a management context, where genotypes may be moved according to seed transfer guidelines it may make more sense to relate analyses based on genetic cluster, but this could also depend on the scale of the spatial distribution.

My second point stems from thinking if this manuscript has an ability to extend insights from genomic offsets beyond its currently application. The authors aptly suggest on Line 327 that their results

support a polygenic model for local climate adaptation and then proceed to take a largely candidate gene approach to assessing genomic vulnerability. This may be the current state of the field, but given the potential impact this paper could have does assessing the RONA for individual genes seem counterintuitive to the fact that climate adaptation for forest trees is polygenic? Are there broader syndromes? Or functional classes of genes that could be assessed to be more informative? I agree that the approach taken here is likely where the field is at – but to push the field forward I wonder if there are alternatives or if the authors at the very least could acknowledge the challenge and point to potential future directions.

Minor Comments:

The introduction had some areas where generalizations were made that could be more specific. Line 43 – no mention of plasticity or adaptive plasticity

Line 47 – are you predicting evolutionary potential or quantifying evolutionary potential (which requires assessment of standing genetic variation)

Line 51 – Reciprocal transplants test for local adaptation – but I'm not sure they test 'capacity' – it's also unclear what 'other approaches' are meant?

Line 57 – What is meant by 'different perspective'?

Line 58 – do you mean only 'future'? or could it be used to assess vulnerability to present – and can you specify 'vulnerability of different population'....(to what?)

Line 63 – what is 'relatively few'? what do we really know here. There is no reference for this statement.

Line 77 – 'along with the characteristics' just remove likely and revise to say forest trees play a major role in the global carbon cycle and are efficient carbon sinks....

Line 81 – can you more specifically relate to how long lifespans, body size, generation length and distribution makes trees particularly vulnerable to maladaptation?

Line 85 – map = mapping

Line 157 – Refer to Table S12

Line 229 – Why $r < 0.6$? It's unclear how variables were identified as there are other BioClim variables that exhibit an $r < 0.6$

Line 308 – remove 'the' in by 'the' heat stress

Line 386 – How does this sentence relate to the genetic capacity to change?

Supp. Fig 22 – what are the color gradients indicating? It is unclear how this figure is interpreted or what it is telling us from a biological perspective as written.

Line 404 – How was genomic load calculated? What databases were used? The details of that analysis were unclear and the assessment of genomic load often is associated with some assumptions so I'm wondering what assumptions were made in that assessment?

Line 412 – the last sentence of this section is confusing – it's unclear how the authors are using the term 'germplasm' – this sentence can be edited for clarity.

Line 440 – plant = plants

Reviewer #2:

Remarks to the Author:

In their manuscript, Sang et al. perform the genome assembly and resequencing of 230 genomes to analyze the landscape genetics and evolutionary history of *Populus koreana*. Using high quality data, the authors identify SNPs, indels and structural variants. They perform genome environment associations and identify loci that could potentially be relevant for future local adaptation. Based on patterns of local adaptation, they project which populations of *P. koreana* could become vulnerable if climate continues to change and they find that in general southern populations would be the most vulnerable.

I think that this is a relevant and interesting manuscript, that will be of general interest for many readers. As the authors mention in the introduction, incorporating local adaptation in projections of

climate change will be necessary for the conservation of species in the future. I also think that the manuscript is very well written and that the methods and results are overall well justified, well explained and well performed. I especially like the evaluation of candidate SNPs using expression data.

Having said that, I do have some general and minor comments that need to be addressed before publication.

General comments:

1) I do not agree with the authors that the adaptive capacity of most species is based on the "species as a whole, but fails to incorporate intraspecific adaptive variation" (L 22-24). While this statement was true a few years ago, there has been a huge increase of studies incorporating local adaptation in projections of climate change (Capblanc et al. 2020 cite around 20 papers using similar methods). Because of this, I do not think that the manuscript is entirely novel. Instead, I think that it is a very nice, and valuable manuscript showing how adaptive diversity is being used to predict vulnerability to climate change. Given this concern, I think that other analyses should be performed before this manuscript can be published in Nature Communications. I suggest a few options below.

2) In conservation genomics, two important aspects that are important before proposing the conservation or management of species, is evaluating both the accumulation of deleterious mutations and the potential outbreeding depression if populations are moved to other locations. I think that both aspects should be evaluated with more detail.

a. First, the authors identified Structural variants (SVs) in their datasets. SVs are known to have a wide distribution of selective effects, going from highly deleterious to highly adaptive (See Zhou et al. 2019, <https://doi.org/10.1038/s41477-019-0507-8>). In this sense, the authors could analyze the accumulation of deleterious SVs in the populations and compare if populations that could be vulnerable in the future, could also have higher proportion of deleterious SVs. The same can be done with SNPs (in lines 402-403; and 592-593, they mention that they have load data). Both Aguirre-Liguori et al. (2021, <https://doi.org/10.1038/s41559-021-01526-9>) and Walvogel et al. (2020) suggest the importance of adding more data to determine how populations will respond to climate change. In this manuscript load is only used to corroborate that GF is not biased by genetic drift.

b. Second, I think it would be very interesting to identify areas where the genetic offset or the RONA of populations would be low if they were moved in the future. For example, Gougherty et al. (2020) used forward offsets to evaluate where the populations would be able to migrate in the future. Also, Rhoné et al. (2020, <https://doi.org/10.1038/s41467-020-19066-4>) identified the migration load of populations to test where they could be moved in the future. Adding this information would allow to identify potential areas where outbreeding depression (or migration load) would be low, if vulnerable populations were actively moved.

3) The distribution of *P. koreana* is highly structured geographically and environmentally. This can increase the identification of outlier loci that are false positives. I think that it would be important to analyze if there are patterns of isolation by environment and determine how strong is the autocorrelation between the geographic and environmental distance (based on PCA) between populations. If the autocorrelation is very strong, maybe it would be convenient to also analyze the vulnerability patterns separating Northern and Southern populations and adding the results in SI.

4) Finally, the text is heavily focused on the potential conservation of *P. koreana*. If conservation is an objective, I think that some caution is needed. First, the authors should make it clear that without experimental validations, their results should be taken with caution. They should also explain what type of experimental validations could be performed. Second, and more important, I think that the authors need to incorporate more future climatic models to incorporate the uncertainty in future models. Here they only use model the model BCC-CSM2-MR model with different shared socio-economic pathways. However, I think that the authors should include more models from Worldclim and that they should show the variation in the estimated offsets and RONA values.

Minor comments.

1) Gene family analyses (Figure 1c): I don't really see the point in showing these analyses. I do not recall seeing anything about the expansion of gene families in the GEA or vulnerability analyses. Perhaps it would be better to use the space for other analyses that I suggested above. Also, could you

please justify the use of outgroups?

2) Lines 64-66: "The processVulnerability". This sentence is not very clear, please explain better what you mean.

3) Lines 120-124: "Repetitive....Table 7). The percentages are not very clear. Can you please explain better?

4) Lines 146-149: I do not understand why comparing Ks shows that the WGD event occurred before divergence. Please explain the logic behind this.

5) Lines 159-161. Can you describe the SFS of indels and SVs? Maybe compare against SNPs

6) Lines 179-181. You mention that FST between groups is weak (0.021), but between pairwise comparisons it can increase above 0.1. I do not think that weak FST between groups this is sufficient to justify that false positives can be low. Especially since you use an FDR of 0.05, instead of Bonferroni corrections at 0.05 to identify outlier loci. That is why, I think that IBE should also be addressed. Also, I would suggest showing the Bonferroni threshold in the Manhattan plots and describe how many SNPs are significant after that correction.

7) Line 214: there are two " - " in genotype-environment.

8) Lines 625: 15 years and mutation rate of 3.75×10^{-8} , do you have a citation for these values? Or a justification?

9) Line 986: transposable elements are almost not considered. I do not think that they are mentioned in GEA or vulnerability analyses. Were they omitted? Why? Should they be also added?

10) Fig 2. Panels a and b should be interchanged. First describe admixture and then the map with the pie plots. Pies are very small, and difficult to see. Can you increase them? Also, I do not know if the selection of colors are the best. Maybe use more contrasting colors that are colorblind friendly. In the map, dark green is -415-300. Can really it really reach values that low below the sea level? Is it extensive geographically areas below sea level? If not, maybe modify the ranks so that it is clearer how the altitudinal distribution goes. It is obvious that there is an environmental barrier between Southern and Northern populations (it is cool!)

11) Figure 3. The threshold line is very weak and difficult to see.

12) Figures 4 & 5. I think I would like to see the RONA and offsets of populations based on different climatic layers, and showing the dispersion across climatic models (maybe used boxplots showing the distribution of summary statistics? For example from Supporting table 17, use the data of one of the important BIOs to show the variation across climatic models).

Reviewer #3:

Remarks to the Author:

Dear editor, dear authors,

In this manuscript, the authors explore the genomic bases of local adaptation to climate in a tree species of northeastern Asia – *Populus koreana* – using a newly assembled reference genome and whole genome resequencing data for 230 individuals. After identifying a set of putative adaptive variants, including SNPs, indels and structural variants, they estimate a risk of maladaptation to climate change across the species range. I found the manuscript interesting and the results mostly convincing, especially the part where the authors explored the characteristics of the loci putatively under selection and found that adaptation to climate may rely more on genetic variations in regulatory regions than in coding regions. Their in-depth analysis of some of the identified adaptive regions (for example in Figure 3) is, to me, the most interesting part of their study, together with the prediction of future maladaptation. However, the narrative and flow of the manuscript could be improved and I have one important concern related to the results of the admixture analysis and other comments and questions that should be addressed before I can recommend this paper for publication in Nature Communications.

My comments are listed below.

Global comments:

1) The introduction section of the manuscript mentions many important ideas, but several statements are erroneous, misleading, or not very well referenced. See my detailed suggestions below.

2) I am surprised by the results of the admixture analysis, which show the presence of two main genetic clusters ($K=2$) that are seemingly not resulting from any geographic or demographic pattern (Figure 2a & 2b). Even more surprising, the genotyped individuals show non-continuous probabilities of assignment to these two clusters, they seem to be "randomly" assigned to 100% of one or the other genetic group, or exactly 50/50 or almost exactly 75/25. This result is not discussed at all by the authors, but I believe it could highlight either some methodological problems (potentially linked to genome assembly or wrongly assembled duplicated regions...) or interesting biological processes (e.g., hybridization, genome duplication...). I think it should be explored and/or at the very least discussed in the study.

3) To me, the phylogenetic analyses (intra- and inter- specific), the inter-species genome comparison (K_s ratio) and the demographic analyses (PSMC) do not seem necessary. They don't bring any critical information to the overall message of the manuscript. Removing these sections would save space to explain other results in more details.

4) I would have appreciated more information about the similarity and/or differences among the three types of markers used in the study. Are indels, SNPs and SVs producing the same results? As it is, the manuscript does not take full advantage of using all three types of genetic markers together.

5) I am not a native English speaker and usually avoid commenting on grammar and phrasing, but the manuscript could benefit from some editing, especially in the Introduction.

Specific comments:

Line 46: I am not convinced that plants have a lower capacity to migrate than most animal species. I would avoid such statement or support it with appropriate references.

Lines 50-52: Is this really the case? To my knowledge, reciprocal transplant experiments / common gardens are used to investigate local adaptation in plant species but not to "assess the capacity for future evolutionary adaptation" as it is stated by the authors. The citation used (Kawecki & Ebert 2004) is a great reference but does not really support the authors' claim.

Line 57: I suggest replacing "different perspective" by "complementary perspective"

Line 63: The statement "relatively few [variants] are expected to be related to climate adaptation" is misleading. Thousands of variants could be located on selected regions, are the authors talking about genes instead of variants? Even so, adaptations to climate can be highly polygenic. Consider revision.

Line 64: Any citation here? Some believe that using the entire genomic information would be more accurate in such predictive models, mainly because identifying all the genetic bases of local adaptation, and its architecture, is very challenging.

Line 77-87: This paragraph is a bit confusing; some ideas are repeated (long lifespan / long-lived, integrating genomic data...) and it is not totally clear to me why a large range and long generation time make trees more sensitive to climate change than other species.

End of introduction: For this last paragraph, I would suggest focusing on the goals and hypotheses of the study more than strictly listing what was done by the authors.

Figure 1:

- I am not sure of what is shown in the center of Figure 1a with all the connecting lines. The caption says "intra-genomic collinear blocks", but these are not really explained anywhere else in the manuscript. The material and methods section makes a difference between "collinear genes" and "syntenic blocks", without enough information to understand the distinction between them, and then talks about "collinear blocks" or "collinear paralogs" (line 145). Are these all the same? More details are needed in the figure caption and the text, even more seeing that the entire genome seems affected by these duplicated regions.
- I find it difficult to interpret/see the distribution curves for (3), (4) and (5) in Figure 1a. Why not using the same plotting strategy as for (1) and (2), which I find much more easier to read.
- We can't really see what is shown in Figure 1b, consider increasing the size of this panel. In the same idea, the overall police size is usually too small.
- I believe panel labels "c" and "d" should be swapped.
- Does 1d still refers to intra-genome syntenic/collinear blocks? It should be mentioned in the caption because showing different species makes it a bit confusing.

Lines 162-170: What about the third genetic cluster that seems to be present in most populations, resulting in a strange pattern of non-continuous assignation within individuals? Could that be due to some methodological problems? Or is there a biological explanation?

Line 216: LFMM means "Latent Factor Mixed Models" and not "latent fixed mixed modeling".

Lines 244-301: Very interesting paragraphs! I believe the main narrative of the study could rely more on these results.

Figure 3:

- Here again the police size is too small, making the scales and labels difficult to read.
- There might be too many results presented on one figure here, it is a bit difficult to get all the information at once. Consider splitting it into two figures or maybe removing some panels (e.g., i and m)

Line 407: The statement may be a bit strong, the only way of being sure of any prediction being to validate it experimentally (Fitzpatrick et al. 2020)

Genomic vulnerability section: This section is interesting but very descriptive of the results and could be improved if interpreted in regards to both the previous findings on adaptive genes and a broader literature.

Material & Methods: I suggest describing all the different parameters used and not just mentioning the abbreviation of the parameters for every program (ex: what does read_cutoff = 1k mean when using the NextDenovo software?)

Lines 551-553: I am not sure what difference the authors make between collinear genes and syntenic blocks. If I'm correct, the same regions are also called collinear paralogs in the manuscript (line 145), I would suggest remaining consistent throughout the document to avoid any confusion. It would also be interesting to know what it means to look for such genomic regions within a single genome compared to different species genomes.

Line 557: What is this evolutionary rate correction?

Line 611: Which genotype matrix was used in this section? The pruned one or the complete one? I'm guessing the complete one but maybe it could be specified to avoid any potential confusion.

Thank you for the opportunity to review this interesting study!

Thibaut Capblancq

Literature cited:

Fitzpatrick, M., Chhatre, V., Soolanayakanahally, R. & Keller, S. (2020). Experimental support for genomic prediction of climate maladaptation using the machine learning approach Gradient Forests, 1–22.

Kawecki, T.J. & Ebert, D. (2004). Conceptual issues in local adaptation. *Ecol. Lett.*, 7, 1225–1241.

Reviewers' comments:

Reviewer #1:

This manuscript represents an impressive amount of effort to use genomic data to test for climate adaptation and quantify populations' vulnerability to climatic change using. Using whole-genome sequences from 230 individuals sampled from across 24 populations the authors combine evaluation of population genomic statistics with environmental association analyses to identify SNPs, indels, and SVs potentially involved in adaptation to climate. They then use this understanding of the genetic basis of contemporary climate adaptation to evaluate genomic vulnerabilities using two metrics, risk of non-adaptedness and genomic offsets to identify regions within the species' range that may be more at risk to changing climatic conditions. Overall, I was extremely impressed with this work and the very careful approach the authors took to data analysis. In general, the manuscript is extremely well-written and clear (but not some parts of the introduction could benefit from clarification) and the methods and figures/tables easy to follow.

Response: We thank the positive comments from the reviewer and appreciate the reviewer's time, patience and constructive comments, which are very helpful for improving the manuscript. As suggested by the reviewer, we have carefully revised the introduction in the current version of the manuscript.

My two main concerns stem from what might be a philosophical consideration, but should be justified in a revision. Specifically (i) understanding when to consider from an analytical and applied conservation perspective genetic clusters independently (as was done for some analyses – but not all) or as representing standing genetic variation for all clusters together across a species' range and (ii) given the degree to which the authors identify and advocate for the polygenic basis of climate adaptation how can data from individual genes (CRL1/HSP60-3A) provide us with respect to risk of non-adaptedness. I elaborate on both below and provide also some minor comments for

revision.

Response: We thank the reviewer for these valuable comments. With regard to the two concerns from the reviewer, we think there might be some misunderstandings here. For the concern (1), we include both northern and southern groups of populations together for all the analyses (including the RONA analyses) in the manuscript as the two clusters likely only diverged recently since they shared large amount of standing genetic variation (as described below). The different colors we used for northern and southern populations in the original Figure 4 may be one factor causing the confusion. In the revised manuscript, we modified the figure and clarified the results. For the concern (2), in the previous manuscript we used the two individual genes (CRL1/HSP60-3A) as examples to show how the RONA was calculated and chose two populations (one from north and one from south) as examples to show the differences of RONA between northern and southern groups of populations for different climatic variables. For each climatic variable, the RONA was indeed calculated independently for each of the putatively adaptive variants associated with the specific climatic variable. The average RONA values across all adaptive variants were previously shown in Figure 4 for two representative variables (BIO5 and BIO13). While, in the current version of manuscript, to exclude the potential confusion resulted from presentation of the figures, we replaced the individual gene examples by the average RONA values calculated across the identified putatively adaptive variants for the 24 populations under two different climatic scenarios (SSP126 and SSP370) when combined with the suggestions from reviewer2.

One of the points I found a little confusing in the manuscript was the fact that Line 204 suggests overall there is weak population structure between southern and northern groups – but the admixture analysis supports a cluster of $K=3$ (Line 164) separating the northern and southern genetic clusters and sometimes they are analyzed together and sometimes apart. To address this, I think first that Figure 2 would really benefit from an understanding of the rangewide distribution of *Populus koreana* – it is unclear if

there is potentially a sampling bias that might be contributing to the identification of genetic clusters and if sampling occurred in the northeastern areas that connect the two regions if a gradient in allele frequencies rather than distinct clusters would be observed. Figure 2C suggests there is likely a pattern of IBD suggesting patterns of post-glacial recolonization, but it was also interesting to note that the authors did not subset the genetic variation used to assess IBD (which could be assessed for neutral loci independently from adaptive loci). Given the lack of overlap in the climatic distribution of the northern and southern clusters (Figure S12) it's a little unclear if populations are exhibiting isolation by distance or potentially isolation by environment (or both). So from Figure S12(d) it seems that there is limited overlap in the environment for the genetic clusters (for BIO1, 2, 4, etc.) so for assessing linear regressions of allele frequencies with environmental variables – does it make sense to model the entire range or would it make sense to model within clusters (a regression for each of the northern and southern clusters). From an applied perspective and thinking how this data may inform conservation priorities and management this may provide a distinct perspective for the relationships of environment with the different clusters. Indeed, for the risk of non-adaptedness (RONA) you model the clusters independently, suggesting that the genetic variation underlying climate adaptation and adaptive capacity of clusters is different. It's also unclear what the black regression line is based on in Figure 4. Given these two distinctions, I think there is opportunity to (a) justify sampling and indicate the entire distribution of *P. koreana* and relate potential biases that may have occurred related to sampling (b) consider whether GEAs should be examined within each genetic cluster or across the species' range as a whole – where components of both approaches have been taken here. From a management context, where genotypes may be moved according to seed transfer guidelines it may make more sense to relate analyses based on genetic cluster, but this could also depend on the scale of the spatial distribution.

Response: We apologize for the confusion with regard to the analyses involved the northern and southern genetic clusters, especially for the reviewer's comments of some

analyses were made by separating the northern and southern genetic clusters apart and some analyses they were analyzed together. This was largely an error in presentation on our part because in our manuscript all analyses, including RONA, were performed including all populations together. Given the reviewer's constructive suggestions, we conducted a few new analyses and also clarified and modified the manuscript in places causing the confusion:

First, we investigated the two-dimensional site frequency spectrum (2D-SFS) of northern and southern genetic clusters. The results highlight that the standing genetic variation are predominantly shared between northern and southern groups of populations, suggesting that the two groups of populations likely diverged very recently and no fixed differences have yet accumulated between the two clusters. When combined with the low genetic divergence observed between the two clusters (both low F_{st} and d_{xy}), analyzing the populations from the two clusters together is likely more appropriate compared to separating them apart. We have added the relevant results in the revised manuscript (Supplementary Figure 7a).

Second, we agree with the reviewer on the point of potential sampling bias may exist in our study, although we think the likelihood may be low. According to the records of “Flora of China” (<http://www.efloras.org>), there are no records of *Populus koreana* distributed in the northeastern areas (which are mainly the regions along the Lesser Khingan Mountains) that connect the northern (Greater Khingan Mountains area) and southern regions (Changbai Mountains area). Also, from our experience of sampling,

we did not find *P. koreana* distributed in those regions. Nevertheless, we cannot exclude the possibility that populations in the intermediate region connecting southern and northern groups might have been destroyed recently by human activities that may have biased our hierarchical structure presented here. According to the reviewer's suggestion, we performed ecological niche modelling using MAXENT v3.3.3 to investigate the current potential range-wide distribution of *P. koreana*, and from the results we find that there is likely some restricted distributions in those regions (the upper-right corner of Figure 2b as shown below). In the revised manuscript, we therefore also discussed the potential influence of sampling bias and the caveat we need to recognize when explaining the results (on line 140-146).

Third, as the reviewer1 and reviewer2 suggested, we performed both the isolation-by-distance (IBD) and isolation-by-environment (IBE) separately for neutral variants (the 535,191 linkage disequilibrium-pruned independent SNPs used for population structure analysis) and adaptive variants (the 1,779 adaptive variants identified by both LFMM and RDA), respectively (Figure 2c and d). From the results, we find that both neutral and adaptive variants display significant patterns of IBD (Figure 2c). After controlling

for the impact of geography, a partial Mantel test was further used to measure and quantify IBE. In contrast to the neutral variants where no significant IBE was observed, the analysis showed strong and significant IBE pattern for adaptive variants (Figure 2d). The same pattern was also found within the southern groups of populations (Supplementary Fig. 6), although not within the northern populations which mostly likely owing to the small number (7) of northern populations that limits the power of the analysis. Moreover, we also performed PCA analysis for the neutral and adaptive variants separately (see the figure below). We found that adaptive structuring (Supplementary Fig. 4b) exhibited a different pattern from neutral genetic clustering (Supplementary Fig. 4a), showing little association with geography and/or population structure. To further decompose the relative contributions of climate, geography and population structure in explaining adaptive and neutral genetic variation, we performed partial RDA and found the exclusive contribution of climate effects explained 41% of the genetic variation of adaptive variants, which was much higher than 10% of neutral variants when controlling for geography and population structure (Supplementary Fig. 4c,d; as shown below). Taken together, all these results suggest that our identification of the environmental-associated adaptive variants when analyzing the northern and southern populations together should be highly robust to the inherent population structure of *P. koreana*. We have added the relevant results and discussion in the revised manuscript, which can be found on line 204-229.

Fourth, we apologize for the confusion of the RONA analyses. In fact, the RONA was assessed using all populations together rather than modelling the northern and southern populations independently. This is largely an issue of the presentation of the results on our side. In the revised manuscript, we modified the Figure 4 according to the comments from all the three reviewers. We incorporated another three future climatic models as suggested by reviewer2, and further integrated the results together to show the average RONA values across the putatively adaptive loci for the 24 sampled populations under two different climate scenarios (SSP126 and SSP370) in 2061-2080 and 2081-2100, respectively.

My second point stems from thinking if this manuscript has an ability to extend insights from genomic offsets beyond its currently application. The authors aptly suggest on Line 327 that their results support a polygenic model for local climate adaptation and then proceed to take a largely candidate gene approach to assessing genomic vulnerability. This may be the current state of the field, but given the potential impact this paper could have does assessing the RONA for individual genes seem counterintuitive to the fact that climate adaptation for forest trees is polygenic? Are there broader syndromes? Or functional classes of genes that could be assessed to be more informative? I agree that the approach taken here is likely where the field is at – but to push the field forward I wonder if there are alternatives or if the authors at the

very least could acknowledge the challenge and point to potential future directions.

Response: We are grateful for the reviewer's suggestions and insights. In the revised manuscript, we extend insights of genomic offsets beyond its current application in two directions. First, in addition to performing genomic vulnerability assessments using both single-loci models of RONA and the gradient forest approach that models a composite genetic turnover function, we further used a recently developed approach to extend the concept of genetic offset by incorporating migration and dispersal into the analysis following Gougherty et al. 2021 as suggested by reviewer2. After measuring the three metrics of local, forward and reverse genetic offsets, the results provide new insights into the assessments of genomic vulnerability under future climate change by simultaneously estimating the contributions of population-level climate maladaptation, minimum migration distances and genotype-climate novelty. The relevant text can be found on line 364-385. Second, in the discussion section, we further add a new paragraph to discuss the limitations and challenges of current assessments of genomic offsets, and also point to the potential solutions and future directions to address these limitations in future studies, which can be found on line 420-444 in the revised manuscript.

Minor Comments:

The introduction had some areas where generalizations were made that could be more specific. Line 43 – no mention of plasticity or adaptive plasticity

Response: We thank the reviewer for pointing this out, and to be more specific, we listed the three ways (movement, plasticity and adaptation) for species to surmount the challenges of climate change, which can be found on line 43-36 in the revised manuscript.

Line 47 – are you predicting evolutionary potential or quantifying evolutionary potential (which requires assessment of standing genetic variation)

Response: We modified the “predicting” as “quantifying” in the revised manuscript.

Line 51 – Reciprocal transplants test for local adaptation – but I’m not sure they test ‘capacity’ – it’s also unclear what ‘other approaches’ are meant?

Response: Thanks to the reviewer for pointing this out. We have revised the whole sentence in the current version of the manuscript (line 53-57).

Line 57 – What is meant by ‘different perspective’?

Response: In the revised manuscript, we have revised the whole sentence, which can be found on line 57-60.

Line 58 – do you mean only ‘future’? or could it be used to assess vulnerability to present – and can you specify ‘vulnerability of different population’....(to what?)

Response: Thanks for pointing this out. We have revised the sentence and also the whole paragraph in the revised manuscript, which can be found on line 52-72.

Line 63 – what is ‘relatively few’? what do we really know here. There is no reference for this statement.

Response: Thanks for pointing this out. We have revised “relatively few” as “only some of these” and also add relevant reference in the revised manuscript, which can be found on line 60-62.

Line 77 – ‘along with the characteristics’ just remove likely and revise to say forest trees play a major role in the global carbon cycle and are efficient carbon sinks....

Response: We have revised the sentence according to the suggestion of the reviewer, which can be found on line 73-74.

Line 81 – can you more specifically relate to how long lifespans, body size, generation length and distribution makes trees particularly vulnerable to maladaptation?

Response: We apologize that the sentence is not clear and accurate in the previous manuscript. We think the main factor causing the trees particularly vulnerable to maladaptation is because of their long generation times, for which the rapid threat of climate change is likely to occur within the lifetimes of individuals, which may further aggravate the consequences of adaptational lag to keep up with rapid climate change. We have modified the sentence in the revised manuscript, which can be found on line 75-78.

Line 85 – map = mapping

Response: Done.

Line 157 – Refer to Table S12

Response: Done.

Line 229 – Why $r < 0.6$? It's unclear how variables were identified as there are other BioClim variables that exhibit an $r < 0.6$

Response: We selected the environmental variables used for redundancy analysis (RDA) mainly considering the ranked importance based on gradient forest analysis and the correlations among these environmental variables (Supplementary Fig. 9a). In addition, we also considered the representativeness of the environmental variables (mainly try to select equal numbers of temperature and precipitation-related variables). After considering these factors, three temperature-related and three precipitation-related environmental variables with pairwise correlation coefficient $|r| < 0.6$ were selected for RDA to avoid overfitting and collinearity. To further evaluate the representativeness of these environmental variables, we retrieved these variables at 60,000 random gridded points across the distribution of *P. koreana*, and used the gradient forest model to predict and map the pattern of climate-associated genomic variation along environmental gradients. By visualizing climate-associated genetic variation across the natural distribution of *P. koreana* (Supplementary Fig. 9b), we found that adaptive genetic variation could be largely explained by these six environmental variables we selected here. In the revised manuscript, we detailed these principles for identifying the uncorrelated environmental variables, which can be found on line 188-197.

We further applied another forward variable selection approach following Capblancq & Forester, 2021, *Methods in Ecology and Evolution*, to select the representative variables, which identify eight environmental variables (BIO2, BIO5, BIO10, BIO13, BIO15, BIO16, BIO18, BIO19) as significantly associated with genetic variation. To further compare the two sets of environmental variables selected in our study and those using the forward selection approach, we performed the partial RDA with the genomic variants (including 5,182,474 SNPs, 736,051 Indels and 30,934 SVs with minor allele frequency higher than 10%). As shown below, we found highly consistent results for the amount of genetic variance explained by all explanatory variables (full model) and by the independent contribution of each set of geography, population structure and climate variables, which again suggest that the six environmental variables chosen in our study should be representative enough to capture the genetic variance that can be explained by environmental variables.

a

Partial RDA models	Inertia	R ² (adj)	p (>F)	PEV	PTV
Full model: $F \sim clim. + geog. + struct.$	90,210	0.577	0.001***	1.00	0.58
Full model: $F \sim clim. (geog. + struct.)$	4,885	0.031	0.069	0.05	0.03
Full model: $F \sim struct. (clim. + geog.)$	38,797	0.248	0.002***	0.43	0.25
Full model: $F \sim geog. (clim.+ struct.)$	1,305	0.008	0.31	0.01	0.01
Confounded climate/structure/geography	45,222			0.51	0.29
Total unexplained	66,054				0.42
Total inertia	156,264				1.00

b

Partial RDA models	Inertia	R ² (adj)	p (>F)	PEV	PTV
Full model: $F \sim clim. + geog. + struct.$	91,068	0.583	0.001***	1.00	0.58
Full model: $F \sim clim. (geog. + struct.)$	5,744	0.037	0.02*	0.06	0.04
Full model: $F \sim struct. (clim. + geog.)$	41,988	0.269	0.001***	0.46	0.27
Full model: $F \sim geog. (clim.+ struct.)$	5,122	0.033	0.016*	0.06	0.03
Confounded climate/structure/geography	38,214			0.42	0.24
Total unexplained	65,196				0.42
Total inertia	156,264				1.00

PEV: The proportion of explainable variance
PTV: The proportion of total variance

*** $p \leq 0.001$; ** $p \leq 0.01$; * $p \leq 0.05$

[a was the results from the partial RDA model with the eight environmental variables (BIO2,5,10,13,15,16,18,19) identified by forward selection approach, b was with the six environmental variables (BIO1,3,5,13,15,19) used in our study]

Line 308 – remove ‘the’ in by ‘the’ heat stress

Response: Done.

Line 386 – How does this sentence relate to the genetic capacity to change?

Response: Thanks for pointing this out. In the revised manuscript, we have added another two parts (three metrics of genetic offsets including local, forward and reverse offsets, and the deleterious genetic load) in the section of “Genomic vulnerability prediction to future climate change” suggested by reviewer2. We thus substantially revised this section and modified this misleading sentence.

Supp. Fig 22 – what are the color gradients indicating? It is unclear how this figure is interpreted or what it is telling us from a biological perspective as written.

Response: Sorry for the unclear interpretation of the figure. It was used for visualization of intraspecific variation in the genotype-climate associations across the distribution of *P. koreana*, where similar colors in the sampled space correspond to similar expected genetic composition at climate-adaptive variants. In general, we retrieved the climate variables at 60,000 random gridded points across the distribution of *P. koreana*, and used the gradient forest model to predict and map the pattern of climate-associated genomic variation along environmental gradients. The genetic turnover was summarized using a principal component analysis (PCA), with the top three components transformed for visualization in a red-green-blue (RGB) color scale. Similar colors in the sampled space correspond to similar expected genetic composition, and loading on the principal components show the direction and magnitude of association with adaptation to different climate variables. We detailed the figure legend in the revised manuscript, which can be found in Supplementary Fig. 9.

Line 404 – How was genomic load calculated? What databases were used? The details of that analysis were unclear and the assessment of genomic load often is associated with some assumptions so I'm wondering what assumptions were made in that assessment?

Response: We apologize for the confusion. In the revised manuscript, we added new metrics for assessing the proxies of genetic load and detailed the analysis. In general, the effects of SNP variants on protein coding sequences were annotated and categorized as loss-of-function (LoF), deleterious (SIFT score < 0.05), tolerated (SIFT score > 0.05), or synonymous based on the Sorting Intolerant From Tolerant (SIFT) algorithm implemented in SIFT4G software using UniRef100 as the protein database. The assumption behind this approach is that the level of conservation in the alignment reflects functional importance, in which the important amino acids will be conserved in the protein family and so changes at well-conserved positions tend to be predicted as deleterious (Ng & Henikoff, 2003, *Nucleic Acids Research*; Vaser et al., 2016, *Nature*

Protocols). The derived versus ancestral allelic state was determined at each SNP position using the est-sfs software through comparison with *P. trichocarpa* sequences. Then, the ratio between the number derived mutations at LoF, deleterious and tolerated sites relative to the number of synonymous variants was calculated and used as proxies for genetic load. We detailed both the methods and the relevant results in the current version of manuscript, which can be found on line 761-778, and on line 386-399, respectively.

Line 412 – the last sentence of this section is confusing – it's unclear how the authors are using the term 'germplasm' – this sentence can be edited for clarity.

Response: We have modified "germplasm" by "genetic resources" in the revised manuscript, which can be found on line 382-385.

Line 440 – plant = plants

Response: Done.

Reviewer #2 (Remarks to the Author):

In their manuscript, Sang et al. perform the genome assembly and resequencing of 230 genomes to analyze the landscape genetics and evolutionary history of *Populus koreana*. Using high quality data, the authors identify SNPs, indels and structural variants. They perform genome environment associations and identify loci that could potentially be relevant for future local adaptation. Based on patterns of local adaptation, they project which populations of *P. koreana* could become vulnerable if climate continues to change and they find that in general southern populations would be the most vulnerable.

I think that this is a relevant and interesting manuscript, that will be of general interest

for many readers. As the authors mention in the introduction, incorporating local adaptation in projections of climate change will be necessary for the conservation of species in the future. I also think that the manuscript is very well written and that the methods and results are overall well justified, well explained and well performed. I especially like the evaluation of candidate SNPs using expression data.

Response: We thank the positive comments from the reviewer and appreciate the reviewer's time, patience and constructive comments.

Having said that, I do have some general and minor comments that need to be addressed before publication.

General comments:

1) I do not agree with the authors that the adaptive capacity of most species is based on the "species as a whole, but fails to incorporate intraspecific adaptive variation" (L 22-24). While this statement was true a few years ago, there has been a huge increase of studies incorporating local adaptation in projections of climate change (Capblanc et al. 2020 cite around 20 papers using similar methods). Because of this, I do not think that the manuscript is entirely novel. Instead, I think that it is a very nice, and valuable manuscript showing how adaptive diversity is being used to predict vulnerability to climate change. Given this concern, I think that other analyses should be performed before this manuscript can be published in Nature Communications. I suggest a few options below.

Response: We agree with the reviewer that there has been an increasing number of studies incorporating local adaptation in projections of climate change. We have modified the sentence in the revised manuscript. In addition, we are grateful for the reviewer's insights and constructive suggestions. We have expanded and performed all analyses suggested by the reviewer, and indeed, we think the manuscript is greatly improved by adding these additional analyses and results.

2) In conservation genomics, two important aspects that are important before proposing the conservation or management of species, is evaluating both the accumulation of deleterious mutations and the potential outbreeding depression if populations are moved to other locations. I think that both aspects should be evaluated with more detail.

Response: We appreciate the reviewer's suggestions and insights. We have added new analyses and results with regard to the two aspects in the revised manuscript, which are detailed as below.

a. First, the authors identified Structural variants (SVs) in their datasets. SVs are known to have a wide distribution of selective effects, going from highly deleterious to highly adaptive (See Zhou et al. 2019, <https://doi.org/10.1038/s41477-019-0507-8>). In this sense, the authors could analyze the accumulation of deleterious SVs in the populations and compare if populations that could be vulnerable in the future, could also have higher proportion of deleterious SVs. The same can be done with SNPs (in lines 402-403; and 592-593, they mention that they have load data). Both Aguirre-Liguori et al. (2021, <https://doi.org/10.1038/s41559-021-01526-9>) and Walvogel et al. (2020) suggest the importance of adding more data to determine how populations will respond to climate change. In this manuscript load is only used to corroborate that GF is not biased by genetic drift.

Response: We thank for the great suggestions from the reviewer. In the revised manuscript, the assessment of genetic load was expanded to include not only the nucleotide diversity, but also the deleterious mutation load estimated based on SNP datasets and also the SV burden across populations. To examine whether the populations with higher genetic offset to future climate change also have an increased burden of deleterious mutation. We first predicted and classified coding SNPs into four categories with respect to their effects using SIFT4G: synonymous, tolerated, deleterious and loss-of-function (LOF). We used the ratios of derived functional

(including tolerated, deleterious or LOF variants) to synonymous variants as proxies for the genetic load. Together, we found no relationship between the predicted genetic offsets (including local, forward and reverse offsets) and both the genetic diversity and the estimated genetic load across populations, even for the LOF variants that are predicted to disrupt gene function and being strongly deleterious (Supplementary Fig. 29a-d, 30a-d). As suggested by the reviewer, we further investigated the associations between the SV burden and the genetic offsets, and similar patterns were observed as that of deleterious mutations, with no relationship between SV burden and genetic offsets being observed (Supplementary Fig. 29e, 30e). As the analysis of genetic offset or the prediction of future climate maladaptation are based on putatively adaptive variations whereas the measures of genetic load depend on the genome-wide distribution of deleterious mutations, we think it might be reasonable to observe the little relationship between them. However, we also discussed that a more thorough understanding of the association between genetic load and the population vulnerability under climate change could benefit from more future studies that incorporate evolutionary processes into the prediction of species' responses to climate change. The relevant results can be found on line 386-399 in the current version of the manuscript.

b. Second, I think it would be very interesting to identify areas where the genetic offset or the RONA of populations would be low if they were moved in the future. For example, Gougherty et al. (2020) used forward offsets to evaluate where the populations would be able to migrate in the future. Also, Rhoné et al. (2020, <https://doi.org/10.1038/s41467-020-19066-4>) identified the migration load of populations to test where they could be moved in the future. Adding this information would allow to identify potential areas where outbreeding depression (or migration load) would be low, if vulnerable populations were actively moved.

Response: We appreciate the reviewer's suggestion and insights. In the current version of manuscript, according to Gougherty et al. (2021) we added another three metrics of genetic offsets (including local, forward and reverse offset) to further integrate

migration to predict potential maladaptation of *P. koreana* to future climate change. We first assessed forward offset with different maximum dispersal distances (100, 250, 500, 1000km and unlimited), and found that although restricting the maximum migration distances unavoidably resulting in higher forward offset, the distribution patterns were largely consistent (Supplementary Fig. 27, 28). We thus estimated the forward genetic offset by identifying the minimum predicted offset assuming that specific contemporary population can migrate to any location in Eurasian continent. Further, after shifting the focus from populations to locations, reverse genetic offset was calculated by identifying the minimum offset for any contemporary population in the current range that best matches the projected future climate of a specific location. Overall, in accordance to the results of RONA and traditional gradient forest-based genetic offset, we found that the southeastern populations nearby the Korean Peninsula were predicted to have relatively high local, forward and reverse offsets (Fig. 5c,d; Supplementary Fig. 24c,d). As such, it is likely that no populations, either locally or elsewhere in the range, existed that are preadapt to future climates in this region. We added the relevant results and discussions in the revised manuscript, which can be found on line 364-385.

3) The distribution of *P. koreana* is highly structured geographically and environmentally. This can increase the identification of outlier loci that are false positives. I think that it would be important to analyze if there are patterns of isolation by environment and determine how strong is the autocorrelation between the geographic and environmental distance (based on PCA) between populations. If the autocorrelation is very strong, maybe it would be convenient to also analyze the vulnerability patterns separating Northern and Southern populations and adding the results in SI.

Response: We are grateful for the reviewer's suggestions and insights. In the revised manuscript, first we used Mantel and partial Mantel tests to assess patterns of isolation-by-distance (IBD) and isolation-by-environment (IBE) separately for the potentially neutral variants (the 535,191 linkage disequilibrium-pruned independent SNPs used for

population structure analysis) and adaptive variants (the 1,779 adaptive variants identified by both LFMM and RDA) (Fig. 2c,d; Supplementary Fig. 6). We found that both adaptive and neutral variants displayed significant patterns of IBD within and between geographic groups, although adaptive variants showed slightly stronger pattern compared to neutral variants (Fig. 2c; Supplementary Fig. 6a,c). However, in contrast to the weak pattern of IBE observed for the neutral variants after controlling for the impact of geography, adaptive variants showed a stronger and significant IBE in partial Mantel tests (Fig. 2d; Supplementary Fig. 6b,d), suggesting that genetic variation of the adaptive variants was mainly influenced by environment. Furthermore, we performed principal component analysis separately for neutral and adaptive variants, and found that different from patterns of neutral genetic clustering, adaptive structuring exhibited little association with geography and/or population structure (Supplementary Fig. 4a,b). Finally, to further decompose the relative contributions of climate, geography and population structure in explaining adaptive and neutral genetic variation, we performed partial RDA and found the exclusive contribution of climate effects explained 41% of the genetic variation of adaptive variants, which was much higher than 10% of neutral variants when controlling for geography and population structure (Supplementary Fig. 4c,d). Taken all these results together and considering that the standing genetic variation are predominantly shared between northern and southern groups of populations from 2D-SFS (Supplementary Fig. 7a), our identification of the environmental-associated adaptive variants when analyzing the northern and southern populations together is supposed to be relatively robust to the confounding effects of population structure and geographical factors, which are expected to be largely shaped by the environmental gradients across the landscape. We have added the relevant results and discussion in the revised manuscript, which can be found on line 204-229.

4) Finally, the text is heavily focused on the potential conservation of *P. koreana*. If conservation is an objective, I think that some caution is needed. First, the authors should make it clear that without experimental validations, their results should be taken with caution. They should also explain what type of experimental validations could be

performed. Second, and more important, I think that the authors need to incorporate more future climatic models to incorporate the uncertainty in future models. Here they only use model the model BCC-CSM2-MR model with different shared socio-economic pathways. However, I think that the authors should include more models from Worldclim and that they should show the variation in the estimated offsets and RONA values.

Response: We appreciate the reviewer's suggestion and insights. First, as the reviewer suggested, we added a new paragraph in the conclusion section in the revised manuscript to specifically discuss the limitations, the experimental validations and other directions for future studies, which can be found on line 420-444 in the current version of manuscript. Second, for the prediction of genomic vulnerability to future climate change, we incorporated another three future climate models in addition to the previous one, including a total of four climate models (BCC-CSM2-MR, ACCESS-CM2, CanESM5 and GISS-E2-1-G model) for all downstream analyses. For the three approaches used to predict the maladaptation to future climates, the RONA and genetic offset metrics were calculated for each of the four climate models and the variation of the estimates were further evaluated, the results of which can be found on Supplementary Fig. 17, 18, 23 and Supplementary Table 16. Given the overall high correlation of the estimates across models, genomic vulnerability inferred by various metrics were presented as the mean across the four climate models. The relevant results can be found in the section of "genomic vulnerability prediction to future climate change" in the revised manuscript.

Minor comments.

1) Gene family analyses (Figure 1c): I don't really see the point in showing these analyses. I do not recall seeing anything about the expansion of gene families in the GEA or vulnerability analyses. Perhaps it would be better to use the space for other analyses that I suggested above. Also, could you please justify the use of outgroups?

Response: In the previous manuscript, we chose four representative species with high-quality genome from the angiosperm phylogenetic tree used as the outgroups. Thanks for the suggestion from the both reviewer2 and reviewer3, we removed these parts in the revised manuscript and save the space for other sections that are more highly associated with the topic of the manuscript.

2) Lines 64-66: “The processVulnerability”. This sentence is not very clear, please explain better what you mean.

Response: Thanks to the reviewer for pointing this out. We have revised the sentence and the whole paragraph in the current version of manuscript, which can be found on line 52-72.

3) Lines 120-124: “Repetitive....Table 7). The percentages are not very clear. Can you please explain better?”

Response: Thanks to the reviewer for pointing this out. We have revised the sentences to make it more clear in the current version of manuscript, which can be found on line 108-110.

4) Lines 146-149: I do not understand why comparing K_s shows that the WGD event occurred before divergence. Please explain the logic behind this.

Response: The K_s value of a pair of homologous sequences, i.e. the estimated number of synonymous substitutions separating them per synonymous site, is used as a proxy for the time elapsed since the sequences diverged. K_s values of paralog pairs can be used to construct a relative age distribution of duplication events in a species, offering insight into the species' gene duplication history. The peaks in such paralog K_s distributions are often used to infer the presence of large-scale duplication events, such as whole genome duplications (WGDs). Ortholog K_s distributions on the other hand are

informative of the age of divergence between species. A common practice to assess the temporal order of speciation and duplication events is to superimpose ortholog and paralog K_s distributions in mixed plot. In our case, all *Populus* and *Salix* species shared the same peak that is older than the other speciation peaks, suggesting that these species shared the same WGD event, which was also certified by multiple previous studies. However, we removed this part in the revised manuscript thanks to the suggestion from the reviewers, as which is not very related to the following main context of the manuscript.

5) Lines 159-161. Can you describe the SFS of indels and SVs? Maybe compare against SNPs

Response: We have added the minor allele site frequency spectrum of SNPs, Indels and SVs in the revised manuscript, which can be found on Supplementary Fig. 2.

6) Lines 179-181. You mention that F_{ST} between groups is weak (0.021), but between pairwise comparisons it can increase above 0.1. I do not think that weak F_{ST} between groups this is sufficient to justify that false positives can be low. Especially since you use an FDR of 0.05, instead of Bonferroni corrections at 0.05 to identify outlier loci. That is why, I think that IBE should also be addressed. Also, I would suggest showing the Bonferroni threshold in the Manhattan plots and describe how many SNPs are significant after that correction.

Response: We thank the reviewer for pointing this out. As shown in our response above, we have addressed the IBE as the reviewer suggested. In addition, we also added the Bonferroni threshold in the Manhattan plots in Fig. 3 and Supplementary Fig. 8, the information of the variants that are significant after the Bonferroni correlation has also been added in the Supplementary Table 13 in the revised manuscript.

7) Line 214: there are two “ - ” in genotype-environment.

Response: We have deleted one in the revised manuscript.

8) Lines 625: 15 years and mutation rate of 3.75×10^{-8} , do you have a citation for these values? Or a justification?

Response: Thanks for pointing this out, and we have added the reference paper in the revised manuscript, which can be found on line 620.

9) Line 986: transposable elements are almost not considered. I do not think that they are mentioned in GEA or vulnerability analyses. Were they omitted? Why? Should they be also added?

Response: Thanks for pointing this out. We have added transposable elements (TEs) and examined the enrichment of adaptive variants located within TEs in the revised manuscript, the relevant results are updated in Supplementary Fig. 12.

10) Fig 2. Panels a and b should be interchanged. First describe admixture and then the map with the pie plots. Pies are very small, and difficult to see. Can you increase them? Also, I do not know if the selection of colors are the best. Maybe use more contrasting colors that are colorblind friendly.

In the map, dark green is -415-300. Can really it really reach values that low below the sea level? Is it extensive geographically areas below sea level? If not, maybe modify the ranks so that it is clearer how the altitudinal distribution goes. It is obvious that there is an environmental barrier between Southern and Northern populations (it is cool!)

Response: We thank the valuable comments and good suggestions from the reviewer. In the revised manuscript, we modified the Fig 2 substantially according to the reviewer's suggestion. We first exchanged the position of admixture and pie plots, and

also increase the size of the pies. In addition, we changed the colors of the plot to make it colorblind friendly, and the ranks of the altitude have also been modified.

11) Figure 3. The threshold line is very weak and difficult to see.

Response: We have modified the figure and made the threshold line more clearly to be read in the revised manuscript.

12) Figures 4 & 5. I think I would like to see the RONA and offsets of populations based on different climatic layers, and showing the dispersion across climatic models (maybe used boxplots showing the distribution of summary statistics? For example from Supporting table 17, use the data of one of the important BIOs to show the variation across climatic models).

Response: We have modified the Figure 4 and Figure 5 to show the RONA and genetic offset metrics under two different climatic layers (SSP126 and SSP370) in the revised manuscript. To show the dispersion of RONA estimates across the climatic models, we chose one temperature-related (BIO5) and one precipitation-related (BIO13) variables as representative variables to show the distribution of RONA estimates across models, which is shown in Supplementary Fig. 17. In addition, the correlations of RONA and genetic offset metrics across the four climatic models at the population level were also calculated and presented in the revised manuscript, the results of which are shown in Supplementary Fig. 18 and Fig. 23.

Reviewer #3 (Remarks to the Author):

Dear editor, dear authors,

In this manuscript, the authors explore the genomic bases of local adaptation to climate

in a tree species of northeastern Asia – *Populus koreana* – using a newly assembled reference genome and whole genome resequencing data for 230 individuals. After identifying a set of putative adaptive variants, including SNPs, indels and structural variants, they estimate a risk of maladaptation to climate change across the species range. I found the manuscript interesting and the results mostly convincing, especially the part where the authors explored the characteristics of the loci putatively under selection and found that adaptation to climate may rely more on genetic variations in regulatory regions than in coding regions. Their in-depth analysis of some of the identified adaptive regions (for example in Figure 3) is, to me, the most interesting part of their study, together with the prediction of future maladaptation. However, the narrative and flow of the manuscript could be improved and I have one important concern related to the results of the admixture analysis and other comments and questions that should be addressed before I can recommend this paper for publication in Nature Communications.

Response: We thank the positive comments of the reviewer, and we appreciate the reviewer's time, patience and constructive comments, which are very helpful. We have revised to improve the narrative and flow of the manuscript, and we also clarified the results of admixture analysis and all other comments raised by the reviewer in the current version of the manuscript.

My comments are listed below.

Global comments:

- 1) The introduction section of the manuscript mentions many important ideas, but several statements are erroneous, misleading, or not very well referenced. See my detailed suggestions below.

Response: We thank the reviewer for pointing this out. In the revised manuscript, we have modified the Introduction substantially to correct the places where the information was previously erroneous and not very well referenced.

2) I am surprised by the results of the admixture analysis, which show the presence of two main genetic clusters ($K=2$) that are seemingly not resulting from any geographic or demographic pattern (Figure 2a & 2b). Even more surprising, the genotyped individuals show non-continuous probabilities of assignment to these two clusters, they seem to be “randomly” assigned to 100% of one or the other genetic group, or exactly 50/50 or almost exactly 75/25. This result is not discussed at all by the authors, but I believe it could highlight either some methodological problems (potentially linked to genome assembly or wrongly assembled duplicated regions...) or interesting biological processes (e.g., hybridization, genome duplication...). I think it should be explored and/or at the very least discussed in the study.

Response: We thank the reviewer for pointing this out, and appreciate the insights from the reviewer. In the revised manuscript, we further performed principal component analysis (PCA) as a complementary method to ADMIXTURE, and found highly consistent results, with the south group of individuals being separated into two clusters on the first PC axis and the northern group being separated from the others on the second PC axis, the other admixed individuals showing intermediate positions between the three genetic clusters (Supplementary Fig. 4a). Considering the high-quality and completeness of the genome assemblies for *P. koreana* and the relatively ancient whole genome duplication events (~58 million years ago) shared by all species of Salicaceae that should have little effects on the genome assembly, we think the most likely explanation is that multiple refugia might exist for *P. koreana* during Quaternary glacial periods. The current distribution of populations is likely to have resulted from postglacial recolonization and secondary contact from these different refugia, especially given that multiple refugia in this area have also been documented in other species, for example the Mongolian oak (Zeng et al., 2015, Molecular Ecology), Asian

butternut (Bai et al., 2016, *New Phytologist*) and two birch species (Wang et al., 2019, *Ecology and Evolution*). In the current version of manuscript, we have explored and discussed the possible evolutionary processes that most likely caused the genetic clustering and population structure pattern in *P. koreana*, which can be found on line 128-146.

In addition, to further explore whether such population structure was caused by some specific genomic regions or was a genome-wide pattern, we used an unsupervised statistical approach called EigenGWAS (Li et al., 2019, *Molecular Ecology*) to identify genomic regions potentially showing “outlier” pattern of population structure through GWAS of eigenvectors. We performed EigenGWAS analysis on the first two eigenvectors (eigenvector 1 and 2) that mostly separates the three genetic clusters. For each eigenvector (ePC1 and ePC2, respectively), we further randomly chose three genomic regions from the variants showing highest, mediate and lowest *P*-values, and found that the PCA plots were highly consistent across the whole genome (as shown below). All these results indicate that population structure and genetic clustering of *P. koreana* is a genome-wide pattern reflecting the shared evolutionary and demographic history, instead of caused by specific genomic regions.

3) To me, the phylogenetic analyses (intra- and inter- specific), the inter-species genome comparison (Ks ratio) and the demographic analyses (PSMC) do not seem necessary. They don't bring any critical information to the overall message of the manuscript. Removing these sections would save space to explain other results in more details.

Response: Thanks to the reviewer for pointing this out. We have removed the phylogenetic analyses and the inter-species genome comparison sections as suggested by the reviewer. In addition, we shorten the PSMC part although keep it since we prefer to associate the analyses with other population structure results to infer the historical demography of *P. koreana* populations.

4) I would have appreciated more information about the similarity and/or differences among the three types of markers used in the study. Are indels, SNPs and SVs producing the same results? As it is, the manuscript does not take full advantage of using all three types of genetic markers together.

Response: We thank the reviewer for pointing this out. In the revised manuscript, we added some new analyses and compared the patterns of SNPs, Indels and SVs in different ways. First, as reviewer suggested, we modified the Fig. 1. Shown in the current figure, the genome-wide distribution as well as the distribution of climatic-associated variants of SNPs, Indels and SVs can be directly compared. In addition, we compared the site frequency spectrum (SFS) of SNPs, Indels and SVs, in which the three types of variants showed highly similar SFS patterns (Supplementary Fig. 2). Moreover, we estimated RONA for the two representative environmental variables (BIO5 and BIO13) across populations independently using the dataset of SNPs, Indels and SVs, which all showed highly consistent results (Supplementary Fig. 22). Lastly, the gradient forest-based estimates of genetic offsets also showed consistent results in

assessing climate change vulnerability through the independent datasets of SNPs, Indels and SVs (Supplementary Fig. 26).

5) I am not a native English speaker and usually avoid commenting on grammar and phrasing, but the manuscript could benefit from some editing, especially in the Introduction.

Response: Thanks for pointing this out. We have modified and edited the manuscript substantially in the revised manuscript, and have taken special care on the editing of the Introduction.

Specific comments:

Line 46: I am not convinced that plants have a lower capacity to migrate than most animal species. I would avoid such statement or support it with appropriate references.

Response: Thanks to the reviewer for pointing this out. We have avoided this statement as the reviewer suggested and also provide relevant references in the revised manuscript, which can be found on line 46-47.

Lines 50-52: Is this really the case? To my knowledge, reciprocal transplant experiments / common gardens are used to investigate local adaptation in plant species but not to “assess the capacity for future evolutionary adaptation” as it is stated by the authors. The citation used (Kawecki & Ebert 2004) is a great reference but does not really support the authors’ claim.

Response: We thank the reviewer for pointing this out, we have corrected both the sentence and the references in the current version of the manuscript, which can be found on line 53-57.

Line 57: I suggest replacing “different perspective” by “complementary perspective”

Response: We have revised the whole sentence in the revised manuscript, which can be found on line 57-60.

Line 63: The statement “relatively few [variants] are expected to be related to climate adaptation” is misleading. Thousands of variants could be located on selected regions, are the authors talking about genes instead of variants? Even so, adaptations to climate can be highly polygenic. Consider revision.

Response: We appreciate the reviewer’s suggestion, and we have modified the statement of “relatively few...” by “only some of these variants...” in the revised manuscript, which can be found on line 60-62 in the current version of manuscript.

Line 64: Any citation here? Some believe that using the entire genomic information would be more accurate in such predictive models, mainly because identifying all the genetic bases of local adaptation, and its architecture, is very challenging.

Response: These are some fair points, and we appreciate the comment. It is truly difficult to identify all the adaptive genetic variants based on current approaches, at the same time there is also possibilities that the signals of local adaptation being overestimated when using the entire genomic information. It seems there is no silver bullet exists to solve the challenges, and we add two relevant references in the revised manuscript, which can be found on line 62.

Line 77-87: This paragraph is a bit confusing; some ideas are repeated (long lifespan / long-lived, integrating genomic data...) and it is not totally clear to me why a large range and long generation time make trees more sensitive to climate change than other species.

Response: Thanks to the reviewer for pointing this out. We have shortened the paragraph in the revised manuscript to remove the repeated information. We think the main factor causing the trees particularly vulnerable to maladaptation is because of their long generation times, for which the rapid threat of climate change is likely to occur within the lifetimes of individuals, which may further aggravate the consequences of adaptational lag to keep up with rapid climate change. We have corrected the part and modified the unclear places in the revised manuscript, which can be found on line 73-78.

End of introduction: For this last paragraph, I would suggest focusing on the goals and hypotheses of the study more than strictly listing what was done by the authors.

Response: We appreciate the reviewers' suggestion. We revised the last paragraph and focused on the three goals that the study aimed to accomplish, which can be found on line 85-91 in the current version of manuscript.

Figure 1:

- I am not sure of what is shown in the center of Figure 1a with all the connecting lines. The caption says "intra-genomic collinear blocks", but these are not really explained anywhere else in the manuscript. The material and methods section makes a difference between "collinear genes" and "syntenic blocks", without enough information to understand the distinction between them, and then talks about "collinear blocks" or "collinear paralogs" (line 145). Are these all the same? More details are needed in the figure caption and the text, even more seeing that the entire genome seems affected by these duplicated regions.

Response: We apologized for the confusion of the figure. Usually a collinear (or syntenic) block was considered to comprise at least five collinear anchor genes. As the species of Salicaceae shared a common ancient whole genome duplication event ~ 58

Mya, most mapped segment of the *Populus* and *Salix* genome had a parallel “paralogous” segment elsewhere in the genome. However, as both reviewer2 and reviewer3 suggested, we removed this part from the revised manuscript and remade the Figure 1.

- I find it difficult to interpret/see the distribution curves for (3), (4) and (5) in Figure 1a. Why not using the same plotting strategy as for (1) and (2), which I find much more easier to read.

Response: We modified the figure following the suggestions of the reviewer.

- We can't really see what is shown in Figure 1b, consider increasing the size of this panel. In the same idea, the overall police size is usually too small.

Response: We moved the Figure 1b to the supplementary figure 1 in the revised manuscript, which is more clearly to read. In addition, following the great suggestion from the reviewer we modified all other figures in the revised manuscript to make them clearer and easier to see.

- I believe panel labels “c” and “d” should be swapped.

Response: We have removed the section and the two figures in the revised manuscript according to the reviewers' suggestions.

- Does 1d still refers to intra-genome syntenic/collinear blocks? It should be mentioned in the caption because showing different species makes it a bit confusing.

Response: The Figure 1d in the previous version of the manuscript referred to the K_s values of pairs of homologous sequences from both paralog and ortholog pairs. K_s values of paralog pairs can be used to construct a relative age distribution of duplication

events in a species, offering insight into the species' gene duplication history. The peaks in such paralog K_s distributions are often used to infer the presence of large-scale duplication events, such as whole genome duplications (WGDs). Ortholog K_s distributions on the other hand are informative of the age of divergence between species. A common practice to assess the temporal order of speciation and duplication events is to superimpose ortholog and paralog K_s distributions in mixed plot. However, we removed this part and also this figure in the revised manuscript thanks to the suggestion from the reviewers, as which is not very related to the following main context of the manuscript.

Lines 162-170: What about the third genetic cluster that seems to be present in most populations, resulting in a strange pattern of non-continuous assignment within individuals? Could that be due to some methodological problems? Or is there a biological explanation?

Response: Considering the high-quality and completeness of the genome assemblies for *P. koreana*, the relatively ancient whole genome duplication events (~58 million years ago) shared by all species of Salicaceae that should have little effects on the genome assembly and read mapping, and the similar patterns of genetic clustering based on the EigenGWAS results throughout the whole genome (as shown in our response above), we think the pattern should not likely caused by methodological issues. Rather, we suggest that the most likely explanation is that multiple refugia (most likely three refugia) might exist for *P. koreana* during Quaternary glacial periods and the current distribution of populations is likely to have resulted from postglacial recolonization and secondary contact from these different refugia, especially given that multiple refugia in this area have also been documented in other species, for example the Mongolian oak (Zeng et al., 2015, Molecular Ecology), Asian butternut (Bai et al., 2016, New Phytologist) and two birch species (Wang et al., 2019, Ecology and Evolution). We have revised and discussed the results in the revised manuscript, which can be found on line 128-146.

Line 216: LFMM means “Latent Factor Mixed Models” and not “latent fixed mixed modeling”.

Response: Thanks for pointing this out, we have corrected it in the revised manuscript.

Lines 244-301: Very interesting paragraphs! I believe the main narrative of the study could rely more on these results.

Response: Thanks for these positive comments.

Figure 3:

- Here again the police size is too small, making the scales and labels difficult to read.

Response: We thank the reviewer for pointing this out. We modified the figure and make the police size larger, at the same time we make the scales and labels easier to read in the current version of the manuscript.

- There might be too many results presented on one figure here, it is a bit difficult to get all the information at once. Consider splitting it into two figures or maybe removing some panels (e.g., i and m)

Response: According to the suggestions from the reviewer, we split the figure and move the panel i and m to the supplementary figure 15 in the revised manuscript.

Line 407: The statement may be a bit strong, the only way of being sure of any prediction being to validate it experimentally (Fitzpatrick et al. 2020)

Response: We have revised it and also added the discussion of the limitation and future directions in the last paragraph of discussion section in the revised manuscript, which can be found on line 420-444 in the revised manuscript.

Genomic vulnerability section: This section is interesting but very descriptive of the results and could be improved if interpreted in regards to both the previous findings on adaptive genes and a broader literature.

Response: We thank the reviewer's comments and insights. In the revised manuscript, as suggested by reviewers, we have expanded the genomic vulnerability section to include a new approach to incorporate migration and dispersal into the assessment and prediction of the putative maladaptation of populations to future climate change (according to the methods developed by Gougherty et al, 2021, Nature Climate Change). In addition, we also associate the estimated genomic vulnerability to future climate change with the proxies of genetic load accumulated across populations, and further evaluate their potential relationships. After incorporating these two main new parts in the section and further modifying the interpretation of the results, we think the section is more interesting and much improved in the revised manuscript.

Material & Methods: I suggest describing all the different parameters used and not just mentioning the abbreviation of the parameters for every program (ex: what does read_cutoff = 1k mean when using the NextDenovo software?)

Response: We thank the suggestion from the reviewer. In the revised manuscript, we described the relevant parameters used for different program, which can be found in the Materials and Methods section in the current version of the manuscript.

Lines 551-553: I am not sure what difference the authors make between collinear genes and syntenic blocks. If I'm correct, the same regions are also called collinear paralogs in the manuscript (line 145), I would suggest remaining consistent throughout the

document to avoid any confusion. It would also be interesting to know what it means to look for such genomic regions within a single genome compared to different species genomes.

Response: We apologize for the confusion. As we responded above, the comparison of these regions within and between genomes could help to assess the temporal order of speciation and duplication events. We have deleted this section according to the suggestion from reviewers since which do not have much connection with the following main parts of the manuscript.

Line 557: What is this evolutionary rate correction?

Response: Because there are differences in evolutionary rates among different species, the K_s peaks of the same whole genome duplication (WGD) may be different, and it is therefore needed to correct the K_s peaks and infer the approximate time of WGD. However, we have removed this part in the revised manuscript as suggested by the reviewers.

Line 611: Which genotype matrix was used in this section? The pruned one or the complete one? I'm guessing the complete one but maybe it could be specified to avoid any potential confusion.

Response: We apologize for the confusion and we used the complete dataset of variants in this section, and we have added the relevant information in the revised manuscript.

Thank you for the opportunity to review this interesting study!

Thibaut Capblancq

Response: Thank you for your great comments and suggestions.

Reviewers' Comments:

Reviewer #2:

Remarks to the Author:

I have reviewed the new version of the manuscript of Sang et al. and I think that the authors have addressed all my concerns and those from the other reviewers. I think this new version of the manuscript has improved considerably, and I think it will be a nice paper for nature communications.

I just have a few comments.

- 1) Most papers use the term genetic or genomic offset, instead of genomic vulnerability. I think it is convenient to keep the common nomenclature. Please change to genomic offset.
- 2) Line 80: please remove "we tried", it sounds like you weren't able to do it!
- 3) I think the manuscript still needs some English editing. There are many little mistakes. However, I think the manuscript is really clear.

Best wishes

Jonás Aguirre

Reviewers' comments to the author:

Reviewer #2 (Remarks to the Author):

I have reviewed the new version of the manuscript of Sang et al. and I think that the authors have addressed all my concerns and those from the other reviewers. I think this new version of the manuscript has improved considerably, and I think it will be a nice paper for nature communications.

Response: We thank the positive comments from the reviewer and appreciate the reviewer's time and patience.

I just have a few comments.

1) Most papers use the term genetic or genomic offset, instead of genomic vulnerability. I think it is convenient to keep the common nomenclature. Please change to genomic offset.

Response: We have modified the "genomic vulnerability" by "genomic offset" in the revised manuscript.

2) Line 80: please remove "we tried", it sounds like you weren't able to do it!

Response: We have removed it in the revised manuscript.

3) I think the manuscript still needs some English editing. There are many little mistakes. However, I think the manuscript is really clear.

Response: We have sent the manuscript to the Genesis Technology Communication (Beijing), Co, Ltd, to let it being edited by native English speaker. In addition, we also checked the manuscript multiple times to correct the little mistakes.